# Local chromatin context regulates the genetic requirements of the heterochromatin spreading reaction

R. A. Greenstein[1,2☯], Henry Ng[1,2☯], Ramon R. Barrales[3¤], Catherine Tan[4,5], Sigurd Braun[3,6], Bassem Al-Sady[1]*

1 Department of Microbiology &Immunology, George Williams Hooper Foundation, University of California San Francisco, San Francisco, California, United States of America, 2 TETRAD graduate program, University of California San Francisco, San Francisco, California, United States of America, 3 Biomedical Center, Department of Physiological Chemistry, Ludwig-Maximilians-Universität of Munich, Planegg-Martinsried, Germany, 4 Department of Cell and Tissue Biology, University of California San Francisco, San Francisco, California, United States of America, 5 Biomedical Sciences graduate program, University of California San Francisco, San Francisco, California, United States of America, 6 Institute for Genetics, Justus-Liebig University Giessen, Giessen, Germany

☯ These authors contributed equally to this work.
¤ Current address: Centro Andaluz de Biología del Desarrollo, Universidad Pablo de Olavide de Sevilla-Consejo Superior de Investigaciones Científicas-Junta de Andalucía, Sevilla, Spain
* bassem.al-sady@ucsf.edu

**Data Availability Statement:** Raw and processed ChIP-seq data is available on GEO (accession GSE184379). Code for flow cytometry, ChIP-seq data analysis, and the primary flow cytometry data

## Abstract

Heterochromatin spreading, the expansion of repressive chromatin structure from sequence-specific nucleation sites, is critical for stable gene silencing. Spreading re-estab-lishes gene-poor constitutive heterochromatin across cell cycles but can also invade gene-rich euchromatin *de novo* to steer cell fate decisions. How chromatin context (i.e. euchro-matic, heterochromatic) or different nucleation pathways influence heterochromatin spreading remains poorly understood. Previously, we developed a single-cell sensor in fission yeast that can separately record heterochromatic gene silencing at nucleation sequences and distal sites. Here we couple our quantitative assay to a genetic screen to identify genes encoding nuclear factors linked to the regulation of heterochromatin nucleation and the distal spreading of gene silencing. We find that mechanisms underlying gene silencing distal to a nucleation site differ by chromatin context. For example, Clr6 histone deacetylase complexes containing the Fkh2 transcription factor are specifically required for heterochromatin spreading at constitutive sites. Fkh2 recruits Clr6 to nucleation-distal chromatin sites in such contexts. In addition, we find that a number of chromatin remodeling complexes antagonize nucleation-distal gene silencing. Our results separate the regulation of heterochromatic gene silencing at nucleation versus distal sites and show that it is controlled by context-dependent mechanisms. The results of our genetic analysis constitute a broad community resource that will support further analysis of the mechanisms underlying the spread of epi-genetic silencing along chromatin.

are deposited in Zenodo (https://doi.org/10.5281/zenodo.6499338). Software version information will be included in conda environment yml files (2021_seqTools.yml for command line data processing and 2021_Renv.yml for analysis via R/Bioconductor).

**Funding:** B.A-S. was supported by grants 1DP2GM123484 and 1R35GM141888 from the National Institutes of Health, grant 2113319 from the National Science Foundation (Division of Molecular and Cellular Biosciences), and a New Frontier Award from the Program for Breakthrough Biomedical Research (partially funded by the Sandler Family foundation). R.A.G. was supported by an ARCS foundation scholarship and a Hooper Graduate Fellowship from the UCSF Department of Microbiology and Immunology. H.N. and C.T. were supported by National Science Foundation Graduate Research Fellowships (grant 1650113). S.B. is a Heisenberg Program Fellow (BR3511/5-1) and was supported by the Network of Excellence EpiGeneSys (HEALTH-2010-257082 funded by the European Commission) and the Collaborative Research Center 1064 (Project-ID 213249687) funded by the Deutsche Forschungsgemeinschaft (German Research Foundation). Flow Cytometry in this work was conducted at the UCSF Flow Cytometry Core which is supported by the UCSF Diabetes Center and grant P30 DK063720 by the National Institutes of Health. Library preparation and genome sequencing was conducted at the UC Davis Genome Center which is partially support by a National Institutes of Health shared instrumentation grant (1S10OD010786-01). The funders had no role in study design, data collection and analysis, decision to publish, or preparation of the manuscript.

**Competing interests:** The authors have declared that no competing interests exist.

## Author summary

Repressive structures, or heterochromatin, are seeded at specific genome sequences and then "spread" to silence nearby chromosomal regions. While much is known about the factors that seed heterochromatin, the genetic requirements for spreading are less clear. We devised a fission yeast single-cell method to examine how gene silencing is propagated by the heterochromatin spreading process specifically. Here we use this platform to ask if specific genes are required for the spreading process and whether the same or different genes direct spreading from different chromosomal seeding sites. We find a significant number of genes that specifically promote or antagonize the heterochromatin spreading process. However, different genes are required to enact spreading from different seeding sites. These results have potential implications for cell fate specification, where genes are newly silenced by heterochromatin spreading from diverse chromosomal sites. In a central finding, we show that the Clr6 protein complex, which removes chromatin marks linked to active genes, associates with the Forkhead 2 transcription factor to promote spreading of silencing structures from seeding sites at numerous chromosomal loci. In contrast, we show that proteins that remodel chromatin antagonize the spreading of gene silencing.

## Introduction

Cellular differentiation requires stabilizing gene expression such that genes coding for lineage-inappropriate information are repressed while genes required for specific cell states are active. Spatial and temporal stabilization of repressive gene states is dependent on the formation and propagation of heterochromatin structures. Heterochromatin is most commonly seeded by DNA sequence-directed nucleation [1,2] and then propagated distally in a sequence-independent process termed spreading to silence genes in neighboring regions. Heterochromatin structures are associated with chromatin marks, such as histone 3 lysine 9 methylation (H3K9me), which are recognized by "readers" that include Heterochromatin Protein 1 (HP1) [3,4]. In some cases, the heterochromatic state then restricts transcription directly through exclusion of RNA polymerase via histone deacetylases (HDACs) [5]. Alternatively, or in parallel, RNA processing pathways promote silencing downstream of heterochromatin assembly [6,7].

The spreading of heterochromatin, and thus gene silencing distal to the nucleation site (nucleation-distal silencing) occurs in at least two very different chromatin contexts: 1. Constitutive heterochromatin, which is generally gene-poor and therefore depleted of activities associated with active genes known to antagonize heterochromatin [8,9]; or 2. heterochromatin involved in regulating cellular differentiation, which is either seeded at new nucleation sites or invades gene-rich euchromatin *de-novo* from existing nucleation sites [10–14]. In either scenario, specific factors may intrinsically promote or antagonize the distal spreading of gene silencing. For constitutive heterochromatin, the inheritance of nucleosomes bearing heterochromatic marks maintains gene silencing across cell divisions [15]. This inheritance promotes modification of nearby nucleosomes through "read-write" positive feedback mechanisms that are intrinsic to heterochromatin histone modifiers [16–18]. In contrast, when heterochromatin invades active chromatin *de novo*, as occurs in differentiation, it will encounter chromatin modifications that can specifically antagonize heterochromatin [8]. Gene silencing at these sites does not benefit from the inheritance of pre-existing marked nucleosomes. Beyond the differences between active and inactive chromatin, it remains unclear whether distinct

nucleation element classes require different regulators to enact efficient spreading outward from those sites.

Fission yeast is an excellent model for addressing the regulation of heterochromatin spreading: 1. It contains a small number of well -defined heterochromatin nucleators, 2. harbors heterochromatin domains that are constitutive as well as others involved in cellular differentiation, and 3. is competent to assemble ectopic heterochromatin domains at nucleation sequences inserted into euchromatin. Over the past four decades, forward and reverse genetic screens in fission yeast have established an exhaustive list of factors required for the nucleation of heterochromatin domains. These nucleation mechanisms include repeat sequences that instruct RNAi-machinery to process noncoding (nc) RNAs involved in targeting the histone methyltransferase Clr4 [19]; signals within nascent transcripts that trigger heterochromatin island formation [20]; pathways involving telomere-protection by the shelterin complex [21,22]; and transcription factor-bound sequences that recruit heterochromatin regulators directly [23]. However, less is understood about factors specifically required for propagating gene silencing through heterochromatin spreading.

We previously developed a fluorescent reporter-based heterochromatin spreading sensor that can assess one key output of heterochromatin (gene silencing) separately from the spatial control of heterochromatin spreading [8,24]. This allows us to address the following questions: 1. Are there known or novel regulators that primarily regulate spreading versus nucleation? 2. Does spreading over chromatin with distinct characteristics, such as gene density or nucleosome arrangements, require different sets of regulators? 3. Do unique heterochromatin nucleation pathways interface with unique heterochromatin spreading regulators? 4. Are there distinct sets of regulators for spreading of the heterochromatin structure and gene silencing? Addressing these questions would elucidate mechanisms that safeguard the genome as well as stabilize specific cell states.

Here, we conduct series of reverse genetic screens in *S. pombe* using a custom collection of gene deletions that target nuclear functions. We investigated gene silencing in different heterochromatin contexts that include several derivatives of the fission yeast mating type (MAT) locus. This gene-poor constitutive heterochromatin region is contained by *IR-L* and *IR-R* boundaries [25–27] and nucleated by at least two elements: (1) *cenH*, which is homologous to pericentromeric *dh* and *dg* elements and relies on ncRNA pathways, including RNAi [26,28,29]; (2) the *REIII* element, a sequence which recruits heterochromatin factors via the stress-response transcription factors Atf1 and Pcr1 [23,30]. We also analyzed an ectopic heterochromatin domain that is embedded in gene-rich euchromatin. This domain is nucleated by an ectopically inserted *dh* element fragment proximal to the *ura4* locus [24,31,32]. Using the combination of MAT derivatives and the ectopic site allowed us to query requirements for nucleation-distal heterochromatin assembly emanating from different classes of nucleators and in different chromatin environments.

Our genetic screen revealed that requirements for heterochromatin spreading differ significantly between distinct chromatin contexts, and to some degree, between different nucleation mechanisms. In particular a specific histone deacetylase (HDAC) complex, Clr6, guided by the Fkh2 transcription factor, is linked to heterochromatin spreading distal to nucleation sites. In contrast, the Clr3 HDAC is generally involved in regulating H3K9me and gene silencing within the entire heterochromatin domain. Our genetic analysis further indicates that there is broad antagonism of heterochromatin-dependent gene silencing at nucleation-distal sites by a diverse set of nucleosome remodelers, in particular Ino80 and Swr1C. Together our genetic analyses allow dissection of both site-specific and broader pathways linked to the spreading of gene-silencing heterochromatin.

## Results

We previously developed a heterochromatin spreading sensor that relies on three transcriptionally-encoded fluorescent protein-coding genes that collectively allow single-cell measurement of heterochromatin formation via flow cytometry, while normalizing for transcriptional and translational noise [24,33]. This method provides separate, quantitative recordings of nucleation-proximal ("green") and distal ("orange") gene expression at a heterochromatin site over large populations of isogenic cells (typically N >20,000, unless strains grow poorly) (**Fig 1A**). The scale of the analysis permits quantitative tracking of unique population distributions, such as multimodal states that would be obscured by ensemble data. When we analyze heterochromatin spreading specifically with our sensor (see Materials and Methods), we do so by examining "orange" in cells that are "green"[OFF], which we take as a proxy for normal nucleation [8,24]. This analysis, therefore, considers the transcriptional consequences of heterochromatin assembly by generating a sensitive readout of the "green" and "orange" reporters that persists for several cell cycles. However, we note that we cannot account for the possibility of highly transient loss-of-nucleation events that do not result in measurably altered transcription at "green".

### Design of heterochromatin spreading sensors that assess four chromatin contexts

To explore whether different genetic contexts utilize general or specific sets of regulators for nucleation-distal gene silencing, we queried three different derivatives of the constitutively heterochromatic mating type (MAT) locus and one euchromatic context, each containing an embedded heterochromatin spreading sensor (**Figs 1A** and **S1**) [24]. The mating-type locus contexts included wild type *MAT*, with the *cenH* and *REIII* nucleating DNA elements intact, and two *MAT* variants that contained mutations in either the *cenH* or *REIII* elements (**Figs 1A** and **S1A** and **S1B**). Mutations in these DNA elements limit initiation of heterochromatin spreading from one nucleator or the other [24]. To probe heterochromatin formation in the euchromatic context, we focused on the *ura4* locus, where heterochromatin spreading is ectopically driven by the upstream insertion of a pericentromeric *dh* DNA element (S1C Fig, [24,31]). We refer to this chromatin context as *ECT* (ectopic).

We first focus on the analysis of strains in which nucleation is driven only by one element, i.e. *MAT ΔcenH* (OFF isolate, see Materials and Methods), *MAT ΔREIII*, and *ECT*. When analyzed by flow cytometry, *MAT ΔcenH* populations appear fully nucleated with near-complete local spreading, as evidenced by population density in the bottom left in the 2D density hexbin plot (**Fig 1E TOP** [24]). *MAT ΔREIII* and *ECT* cell populations, while mostly nucleated, display a stochastic distribution of spreading states, as evidenced by a vertical distribution on the left of the 2D density histogram (**Fig 1G and 1I TOP** [24]). While the distance between nucleation and sensor sites varies slightly for the different chromatin contexts analyzed (from 2.4 to 3.6 kb; see S1 Fig), we showed previously that altering the distance between "green" and "orange" does not qualitatively affect the output [24]. Thus, we presume that these differences in behavior reflect different intrinsic properties of the chromatin environment rather than the difference in spatial distribution relative to the nucleation site. In addition to the wild-type background, we assessed *Δclr3* as a reference point for strong loss of gene silencing (**Fig 1E, 1G and 1I, MIDDLE**).

### Identification of chromatin context-specific positive spreading regulators

In order to identify nuclear factors linked to context-specific spreading, we crossed a deletion library of ~400 nuclear function genes (see Materials and Methods and **S1 Table** and **Fig 1B**)

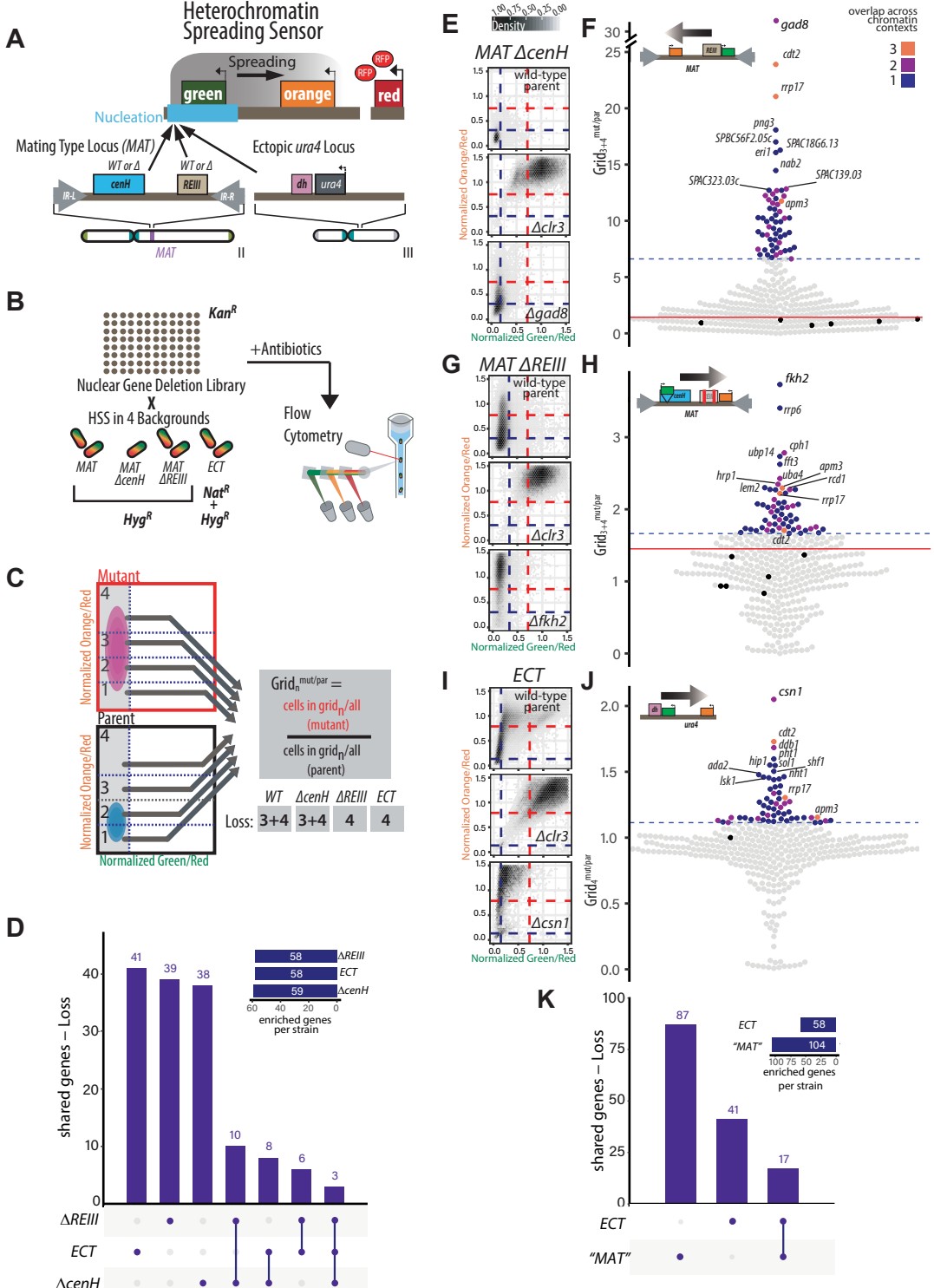

**Fig 1. A genetic screen based on a suite of fluorescent reporters identifies regulators of heterochromatin nucleation-distal gene silencing in different chromatin contexts. A.** TOP: Overview of heterochromatin spreading sensor (HSS, [24]). Three transcriptionally encoded fluorescent protein genes are integrated into the genome. *SFGFP* ("green") proximal or internal to the nucleation site allows identification of heterochromatin nucleation; *mKO2* ("orange") distal to the nucleation site allows identification of heterochromatin spreading. *3xE2C* ("red") in a euchromatin region normalizes cell-to-cell noise. BOTTOM: The endogenous mating type locus (*MAT*) and heterochromatin ectopically targeted to the *ura4* locus [24] were examined with

the HSS in the screen. In *bona fide* mutations of the *MAT* nucleators, *cenH* and *REIII*, nucleation is only initiated at one site. See S1 Fig for detailed diagrams. **B.** Workflow of the screen to identify genes that contribute to nucleation-distal gene silencing. A custom nuclear function deletion library (S1 Table) was mated with four different reporter strains (*WT MAT*, *MAT ΔcenH*, *MAT ΔREIII and ECT*). The fluorescence of "green", "orange" and "red" for each mutant cell within each background are recorded by flow cytometry. **C.** Overview of the loss-of-spreading analysis with mock distributions of cells and grids indicated. To identify cells that have successfully nucleated heterochromatin and are silencing-competent, "green"-off populations (successful nucleation events) are isolated first. Within these populations, enrichments of cell populations in particular "orange" fluorescence ranges (Grid$_n$) are calculated as Grid$_n^{mut/par}$. This second step specifically assesses for any loss of nucleation-distal gene silencing. As an example, In *WT MAT* Grid$_{3+4}^{mut/par}$ is calculated as percentage of the mutant population divided by percentage of parent population in Grid$_{3+4}$. The Grids (either 3+4 or 4) used for analysis of loss of spreading in the four chromatin contexts are indicated. **D.** Upset plots indicating the frequency of "loss of spreading" gene hits appearing in one or multiple singly nucleated chromatin contexts. For each bar, the chromatin context(s) with shared phenotypes for the underlying gene hits is indicated below the plot. The inset indicates the total number gene hits for loss of spreading in each chromatin context. "Shared genes": number of genes that appear as "loss of spreading" hits across the number of indicated chromatin contexts. **E.** *MAT ΔcenH* 2D-density hexbin plots of the wild-type parent, a strong heterochromatin loss hit (*Δclr3*), and the top loss of nucleation-distal silencing hit (*Δgad8*) in this chromatin context. Dashed blue lines indicate the values for repressed fluorescence state and dashed red lines indicate values for fully expressed fluorescence state. **F.** Beeswarm plots of Grid$_{3+4}^{mut/par}$ for *MAT ΔcenH* loss of nucleation-distal silencing hits and number of overlapping contexts. The top 10 hits are all annotated, and below those hits, mutants that show overlap with at least 2 other chromatin contexts are additionally annotated. Red line, 2SD above the Grid$_{3+4}^{mut/par}$ of the wild-type parent isolates (black dots); dashed brown line, the 85th percentile; Dot color, number of chromatin contexts with loss of spreading phenotype over the cutoff. This mutation allows examination of only the *REIII* nucleation site at the *MAT* locus. **G.-H.** Beeswarm plots of Grid$_4^{mut/par}$ for the *MAT ΔREIII* strain were analyzed and displayed as in E. and F. This mutation allows examination of only the *cenH* nucleation site at the *MAT* locus. **I.-J.** Data for the *ECT* strain were analyzed and displayed as in G. and H. This strain examines a euchromatic context and analyzes silencing of at the *ura4* locus by spreading from an ectopic nucleation site. **K.** Upset plots indicating the frequency of "loss of spreading" gene hits appearing in singly nucleated MAT contexts (*MAT ΔREIII* and MAT *ΔcenH*) versus the *ECT* context.

to the four reporter strains above and measured nucleation and spreading by flow cytometry. To segregate proximal from distal silencing, we first isolated cell populations that reside within a "green"-off gate, which represents cells with heterochromatin fully assembled at the nucleation site and no expression of the reporter (see Materials and Methods, **Fig 1C** and [24]). Within this gate, we divide the "orange" signal into 4 grids, from fully repressed (i.e., complete spreading over "orange"; Grid 1) to fully de-repressed (i.e., no silencing at "orange"; Grid 4), with the remaining space symmetrically divided to yield Grids 2&3 (see Materials and Methods). To quantify increased or decreased nucleation-distal silencing in a given mutant, we calculated a Grid$_n^{mut/par}$ metric (described in Materials and Methods), which tracks the changes of cell distributions in "orange" expression within the "green"-off gate (**Fig 1C**). Since *MAT ΔcenH* and also *WT MAT* display very tight silencing of both "green" and "orange" with very few events in grid 4 (**Figs 1E** and **S2A**), we used a Grid$_n^{mut/par}$ metric that considers both grids 3+4 for the robust identification of spreading defects in these strain backgrounds. *MAT ΔREIII* and *ECT* have a more stochastic spreading behavior with "green"-off cells populating a range of "orange" states from OFF to ON [24], including, to some extent, both grid 3 and 4 (**Fig 1G** and **1I**). Hence, for these two contexts we used a Grid$_n^{mut/par}$ metric that only considers grid 4 to focus on complete loss of spreading (i.e. "orange" signal in the range of *Δclr4* control). We used two metrics to define gain or loss of spreading mutants for further analysis. The first was a significance threshold used if multiple parental isolates were available (all except *ECT*). To meet this criteria Grid$_n^{mut/par}$ values had to be at least 2 standard deviations (2SD) above the mean of the parent isolates (red line, **Figs 1F** and **1H** and **S2C, S2D, S2F and S2H**). As an additional cut-off, we only considered the top 15% of all Grid$_n^{mut/par}$-ranked mutants, even though more genes passed the 2SD significance threshold, primarily to focus on the genes with the highest impact on nucleation-distal silencing (blue dotted line). Having identified these gene hits, we proceeded to analyze their relationships within and across chromatin contexts.

As general note on how we describe the function of our screen hits, in cases where we can correlate silencing defects at the spreading reporter to altered H3K9 methylation, we refer to these changes as heterochromatin spreading defects. In other cases where we lack information on H3K9 methylation, we refer to these changes as nucleation-distal gene silencing defects, as it is in principle possible that certain mutants can affect gene silencing without affecting structural features of heterochromatin.

We first examined the degree to which modulators of nucleation-distal silencing are shared between chromatin contexts where heterochromatin spreading is driven by one major element, via upset plots (**Figs 1D** and **S2I** for all four contexts). Conceptually similar to a Venn diagram, this analysis allows rapid visualization of the degree of overlap between data sets, with the number of shared hits plotted as a bar graph and the sets each bar represents annotated below the plot. For loss of nucleation-distal silencing (*i.e.* mutants in genes that promote spreading, **Fig 1D**), these upset plots showed that exceedingly few genes were shared across all chromatin contexts (i.e. 3 out of 145 unique hits for singly-nucleated contexts, Fig 1D; 2 of 164 unique for all contexts hits, **S2I Fig**). Notably, two of these three genes (*apm3*, *rrp17*) have not previously been implicated in heterochromatin assembly. Apm3 has been proposed to be part of an AP-3 adaptor complex that mediates vesicle trafficking, whereas Rrp17 is a predicted rRNA exonuclease.

We expanded this analysis to also include the *WT-MAT* context. Six genes were shared across all the MAT locus chromatin contexts (*apl5*, *cph1*, *hrp1*, *spt2*, *snt2*, *pcf1*). In contrast, the majority of hits (i.e. 101 genes for all contexts, 118 when examining singly nucleated contexts) were linked to regulation in only one chromatin context. The degree to which genes contributed positively towards spreading ($Grid_n^{mut/par}$) and the degree of overlap across chromatin contexts (by color code) is shown in **Figs 1F**, **1H**, **1J** and **S2B**, **S2D**, **S2F** and **S2H** along with the top loss-of-spreading hit for each context (**Figs 1E**, **1G**, **1I** **BOTTOM** and **S2A** **BOTTOM**). A similar picture emerged on a more coarse-grained level, comparing the two major chromatin environments, MAT and ECT. To do so we grouped both singly nucleated MAT contexts (*MAT ΔREIII* and *MAT ΔcenH*) and compare them to *ECT*. Even in this small comparison of two groups, only 17 genes are shared out of a total of 145 (**Fig 1K**). The low overlap of genes shared across specific chromatin contexts and environments in the genome emphasizes that specific contexts have a strong impact on nucleation-distal silencing.

Examining the top hits, we make the following observations: Pathways that restrain the heterochromatin antagonist Epe1 play a prominent role in promoting nucleation-distal silencing. Both the COP9 signalosome and an E3 ubiquitin ligase complex comprising the adaptor proteins Ddb1 and Cdt2 are involved in Epe1 turnover [34,35]. *cdt2* is one of the three hits conserved across all singly nucleated contexts. We found that *csn1* (COP9), *cdt2*, and *ddb1* are the three strongest hits in *ECT* and that *cdt2* is also among the top hits in *MAT ΔcenH*. The histone chaperone, Asf/HIRA (*hip1*) and histone variant H2A.Z (*pht1*) were top hits linked specifically to *ECT* regulation (**Fig 1J**). H2A.Z was just recently shown to play a role in maintaining RNAi-driven heterochromatin in *S. pombe* [36] and is known to antagonize heterochromatin spreading in budding yeast [37], indicating that a role for this histone in heterochromatin domain expansion control is conserved even though it functions in opposite directions in the two systems. The transcription factor gene *fkh2* was a top hit in regulation of *WT MAT* and *MAT ΔREIII*, along with *rrp6*, a key member of the nuclear exosome (see below; **Figs 1G** and **S2A**). We also saw strong hits that were unique to *MAT ΔcenH*. The top hit here was *gad8*, which encodes a protein kinase that targets several factors, including Tor1 and Fkh2 (**Fig 1E**). In short, specific gene sets seem to regulate spreading distal to the nucleation site in different chromatin contexts. Though specific, some of these genes play a conserved role in regulating chromatin spreading in other organisms.

As mentioned above, loss of the AP-3 adaptor complex subunits Apm3 and Apl5 induced nucleation-distal gene silencing, and as well show below, impairs H3K9me2 accumulation. Interestingly, loss of Apm3 affected spreading in all chromatin contexts, whereas Apl5 affected only the *MAT* contexts (also observed in [38]). We further assessed the activities of Apm3 and Apl5 by generating *Δapm3* and *Δapl5* single and double mutants in the *MAT ΔREIII* background, where *Δapm3* and *Δapl5* had a moderate and mild effect in the screen, respectively. We reproduced the mild to moderate spreading defect for both single mutants; we further observed a slightly stronger defect in the *Δapm3 Δapl5* double mutant (Grid$_4$$^{mut/par}$ 1.56) compared to the *Δapm3* and *Δapl5* single mutants (Grid$_4$$^{mut/par}$ 1.4 and 1.16, respectively); **S3A–S3D Fig**). Whereas Apl5 is largely cytoplasmic, thus likely acting indirectly, Apm3 shows both nuclear and cytoplasmic localization, (**S3E and S3F Fig**) and also affects H3K9me2 accumulation at heterochromatin islands (**S3G Fig**). Together, these findings suggest a direct rather than indirect role for Apm3 in heterochromatin assembly. However, further work is needed to elucidate the specific function of Apm3 in heterochromatin spreading and whether this is linked to the AP-3 complex itself.

## Identification of negative spreading regulators

In addition to loss of nucleation-distal silencing (positive regulators), we also identified mutants that showed gain of silencing (negative regulators). We examined a Grid$_n$$^{mut/par}$ metric that considers grid 1, as increased distribution into grid 1 indicates silencing of "orange" beyond the wild-type. We could not examine *MAT ΔcenH* for this phenotype because this chromatin context is highly repressed in the OFF state as reported previously and the vast majority of cells are already resident in grid 1(**Fig 1E**) [24,39]. As before, we examined first singly nucleated contexts in which gain of spreading can easily be detected, i.e. *MAT ΔREIII* and *ECT* (**Fig 2**) and separately also all three contexts including *WT MAT* (**S4A–S4F Fig**). Even though *ECT* displays a very similar spreading behavior to *MAT ΔREIII* (24), we found little overlap between the two, with only 10 genes shared between them (for *MAT ΔREIII* and *ECT*: 10 shared out of 92, **Fig 2B**; examples for top hits are shown in **Fig 2D and 2F**. For all contexts including *WT MAT*: 5 shared out of 98, these are *vps71*, *arp6*, *leo1*, *git1*, and *pmk1*, **S4G Fig**). Of the more limited number of genes shared across all three contexts, we note that Leo1 was previously shown to be implicated in spreading control across boundaries [40], whereas Vps71 and Arp6 are members of the H2A.Z-specific Swr1 remodeling complex [41]. The larger number of antagonists unique to *ECT* (**Figs 2B** and **S4G**) suggests that heterochromatin spreading is under additional layers of control in the euchromatic context.

We sought to independently validate the above observations. We selected 5 mutants that have chromatin context-specific effects, covering both loss and gain of spreading: *saf5*, which shows gain of spreading in *WT MAT* and mildly in *MAT ΔREIII*; *eaf6*, which shows gain of spreading in *MAT ΔREIII* only; *pht1* and *hip1*, which show loss of spreading only in *ECT;* and *gad8*, which shows loss of spreading primarily in *MAT ΔcenH*. We recreated these mutations *de novo* in the chromatin contexts described above and conducted RT-qPCR analysis for *SF-GFP* ("green", nucleation) and *mKO2* ("orange", spreading) transcripts. This validation approach broadly recapitulated our initial screen (**S5 Fig**), confirming that these gene products play context-specific functions in nucleation-distal silencing.

## Chromatin remodelers broadly antagonize nucleation-distal silencing

Taking our analysis beyond individual genes, we sought to query which protein complexes are involved in regulating nucleation-distal gene silencing, as this may highlight the major pathways involved in this process. Using the Gene Ontology (GO) protein complex annotations

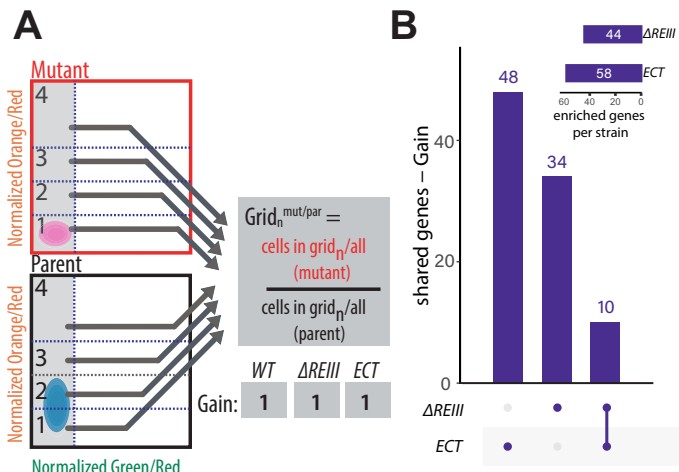

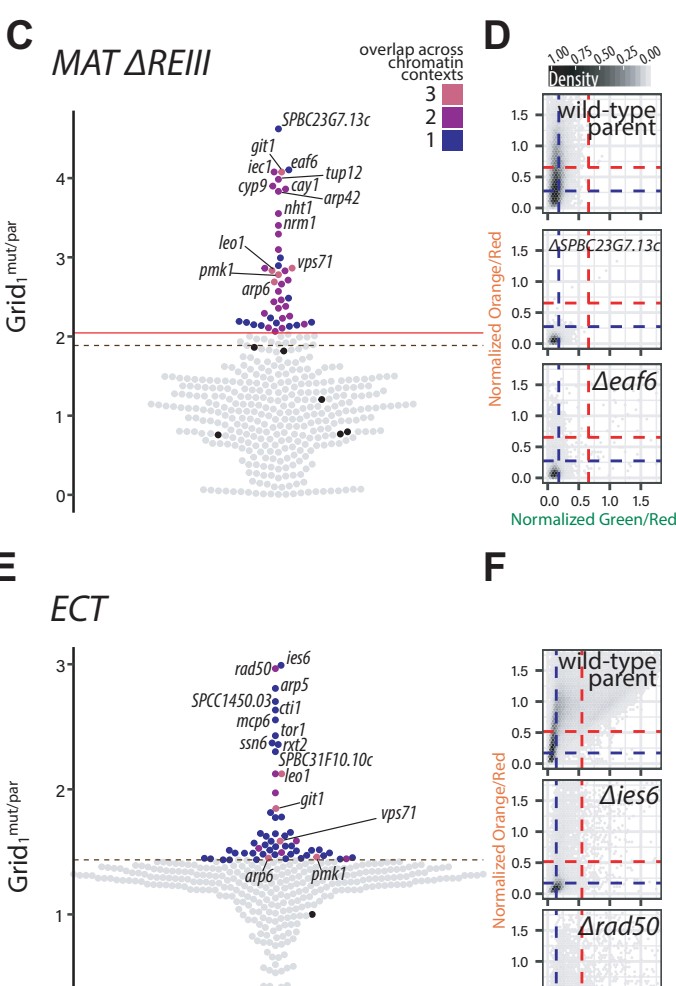

**Fig 2. Gain of nucleation-distal gene silencing in MAT *ΔREIII* and *ECT* chromatin contexts. A.** Overview of the gain of nucleation-distal silencing analysis with mock distributions of cells and grids indicated, as in Fig 1C. To identify gain of spreading mutants in *ECT* and *MAT ΔREIII*, $Grid_1^{mut/par}$ was calculated as percentage of the mutant population divided by percentage of parent population in $Grid_1$. **B.** Upset plots indicating the frequency of Gain of Spreading gene hits that appear in one or both chromatin contexts as in Fig 1D. For each bar, the chromatin context(s) with shared phenotypes for the underlying gene hits is indicated below the plot. The inset indicates the total number gene hits in each chromatin context of the same phenotype. **C.** Beeswarm plots of $Grid_1^{mut/par}$ for *MAT ΔREIII* gain of nucleation-distal silencing hits. The top 10 hits are all annotated, and below those hits, mutants that show overlap with 2 other (*WT MAT* and *ECT*) chromatin contexts are additionally annotated. Red line, 2SD above the $Grid_1^{mut/par}$ of wild-type parent isolates (black dots); dashed brown line, the 85th percentile; Dot color, number of chromatin contexts with loss of spreading phenotype over the cutoff. **D.** *MAT ΔREIII* 2D-density hexbin plots of the wild-type parent, and the two top gain of spreading hit of this chromatin context. Dashed blue lines indicate the values for repressed fluorescence state and dashed red lines indicate values for fully expressed fluorescence state. **E., F.** As in A. and B., but for *ECT*.

from Pombase [42], we annotated each gene hit that met the criteria for further analysis as outlined above for all four chromatin contexts. We then tabulated the frequency ("counts") of each GO complex by chromatin context for both loss of distal silencing (loss) and gain of distal silencing (gain) phenotypes, performed unsupervised clustering on the data, and displayed the results as a heatmap (Fig 3). Overall, we identified three major common trends: 1. A broad role for chromatin remodelers in antagonizing nucleation-distal silencing; 2. a role for the SAGA complex in promoting nucleation-distal silencing at *ECT*; 3. a role for Clr6 histone deacetylase complexes (HDACs) in promoting nucleation-distal silencing at MAT, with the notable exception of the Set3C module (part of expanded Rpd3L complex), which antagonizes distal silencing.

Mutants defective in chromatin remodeling complexes strongly contribute to the "gain" phenotypic category, which includes the Swr1C, Ino80, SWI/SNF, and RSC-type complexes (Fig 3). To explore this further, we assessed which protein components contributed to these GO complex counts. For all genes annotated to a given chromatin remodeling complex and present in our screens, we displayed whether they were identified as a hit (blue) or not (grey) in a hit table (Fig 4A). Indeed, we found that the large majority of the gene hits annotated fall within the "gain" but not "loss" phenotype across backgrounds, confirming that these nucleosome remodeling complexes potentially antagonize spreading (see examples 2D hexbin plots, Fig 4B).

## SAGA primarily promotes heterochromatin spreading in the euchromatic context

A surprising observation was the enrichment of a large number of subunits of the SAGA complex among the loss-of-nucleation-distal silencing hits (Fig 3). Indeed, six SAGA subunit genes were associated with altered expression of the *ECT* reporter, the most enriched single complex for any reporter strain (Figs 3, and 4C). This suggests that SAGA, a histone acetylase involved in gene transcription, positively regulates nucleation-distal silencing in euchromatin (see example 2D hexbin plots, Fig 4D). To assess if SAGA directly influences heterochromatin spreading, rather than indirectly affecting gene silencing, we assessed H3K9me2 accumulation at "green" and "orange" in the *ECT* background, as well as the pericentromeric *dg* element. Consistent with the results of the genetic screen, we find that the histone acetylase catalytic subunit Gcn5 is required for efficient spreading of H3K9me2 to "orange", but not its establishment at "green" or at *dg* (S6 Fig).

## Clr6 HDAC complexes promote nucleation distal silencing, primarily in constitutive heterochromatin

Three classes of HDACs exist, which have both redundant and non-overlapping functions in the formation of heterochromatin domains and gene silencing. Clr6 belongs to class I and is

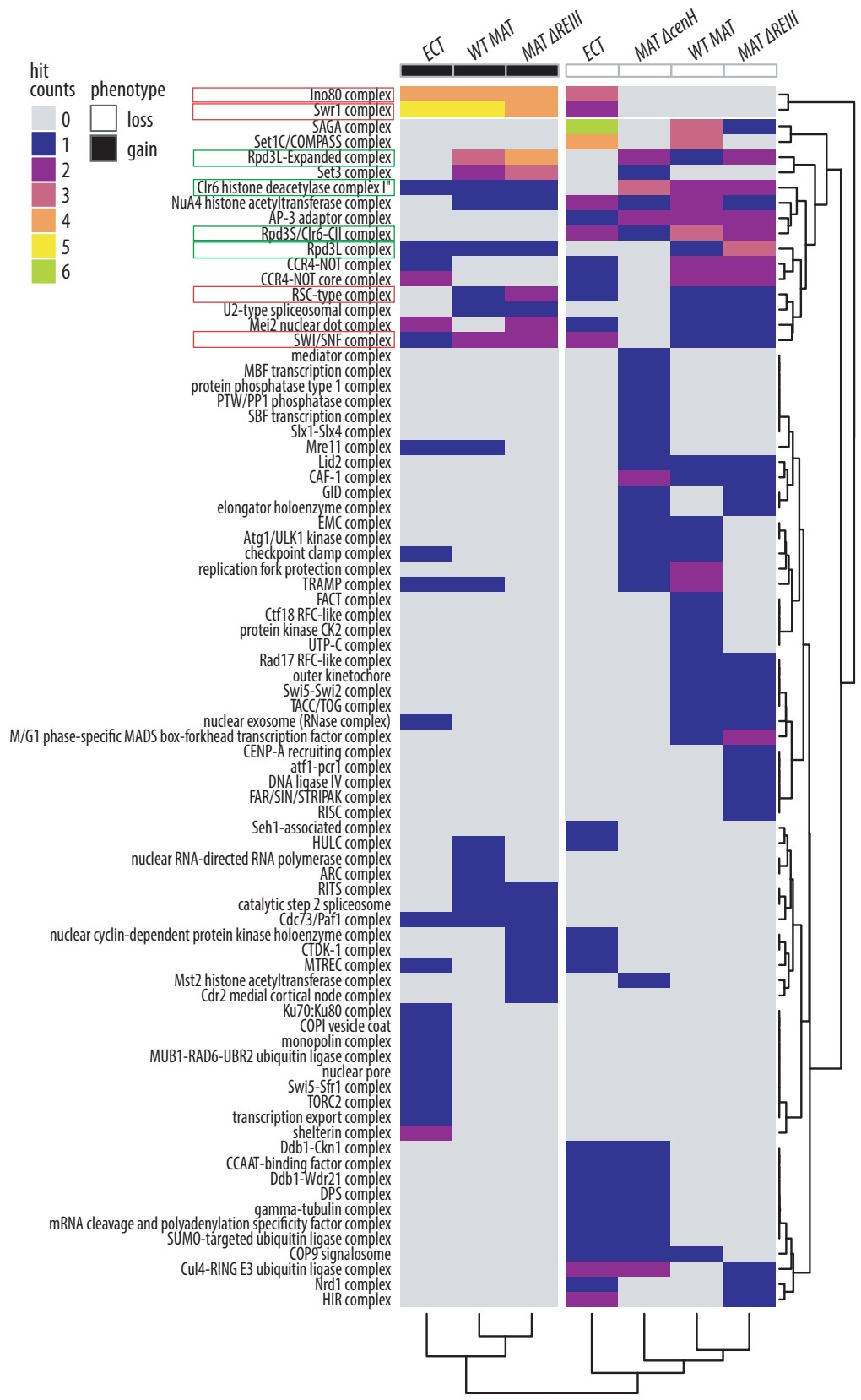

**Fig 3. Heterochromatin nucleation-distal gene silencing is regulated by sets of unique and common protein complexes across different chromatin contexts.** Heatmap of GO complex annotations for hits in each category and strain. Rows, representing GO complexes annotated to genes within the screen that were identified as hits, are arranged via hierarchical clustering. Columns are defined by the hit phenotype (loss of nucleation-distal silencing - white; gain of nucleation-distal silencing - black), and each screen chromatin context is indicated at the top. The columns were clustered by hierarchical clustering and the dendrogram was cut to define 2 branches. Red boxes, chromatin remodeling complexes; Green boxes, Clr6 complexes (Note that Rpd3L Expanded includes Set3C).

part of several sub-complexes, contributing to both heterochromatic and euchromatic gene regulation [43,44]. Clr3 belongs to class II and is a member of the SHREC complex [45], whereas Sir2 is a class III HDAC of the sirtuin family [46]. Based on our screen, class II and III HDACs affect nucleation and distal gene silencing equally, indicating that they do not act in a spreading-specific manner. (Clr3, **Fig 1**; Sir2, **S7 Fig**). In contrast, sub-complexes of the Clr6 family, including Rpd3S, Rpd3L, and Clr6 I″, contribute exclusively to nucleation-distal but not proximal silencing (hit table, **Fig 4E;** 2D hexbin plots, **Figs 4F** and **S8**). As noted above, the forkhead transcription factor Fkh2 was identified amongst the strongest nucleation-distal silencing hit in *WT MAT* and *MAT ΔREIII* reporter strains. Despite not being formally annotated to the Clr6 I″ complex by GO terms, Fkh2 has previously been linked to this sub-complex [47]. Based on this previous analysis, and our genetic data, we considered Fkh2 to be a member of Clr6 I″. While Rpd3L/ Clr6 I′, Clr6 I″, and Clr6S (Complex II) positively contributed to spreading ("loss" phenotype), several members of the Rpd3L-Expanded complex antagonized spreading and were found as hits inducing a "gain" phenotype (**Figs 4E**, **4F** and **S8**). This includes a subset belonging to the Set3 Complex (Set3, Hif2, Hos2, Snt1). We validated our analysis by examining the phenotype of the *de novo* generated gene deletion strains of *fkh2* and *prw1*, which encodes a core structural subunit of the Clr6 HDAC complexes, in the *MAT ΔREIII* heterochromatin spreading sensor (**Fig 4F, bottom**). The *Δfkh2 Δprw1* double mutant displayed a similar phenotype to the *Δprw1* single mutant, corroborating the hypothesis that Fkh2 is part of the same complex as Prw1 (**Fig 4F, bottom**).

Overall, these data suggest that Clr6 I′, Clr6 I″ and Clr6S HDAC complexes specifically promote nucleation-distal heterochromatic gene silencing.

## Clr6 complexes promote distal H3K9 methylation spreading at telomeres, pericentromeres and islands

The analysis above suggests that Clr6 complex subunits contribute to the spreading of gene silencing. We therefore examined the mechanisms underlying this phenotype and examined the role of Clr6 relative to another prominent heterochromatic HDAC, Clr3, which is well-established as a key regulator heterochromatin in gene silencing [45,48]. First, we examined whether the phenotype observed for *Δprw1* and *Δfkh2* holds true for *clr6* itself, which encodes the catalytic subunit in the complex. We tested the phenotype of the *clr6-1* allele, which has a hypomorphic phenotype at permissive temperatures (note that *clr6* is an essential gene) [43]. Under these conditions, *clr6-1* shows moderate spreading defects in *MAT ΔREIII* without affecting nucleation, which is consistent with its role in nucleation-distal silencing (**Fig 5B** vs. **5A**). In contrast, a catalytically dead mutant of Clr3 (*clr3-D232N*) showed complete loss of silencing (**Fig 5C** vs **5A** and **Fig 1**). This complete loss of silencing is identical to the null *Δclr3* (**Fig 1**), supporting further analysis using the hypomorphic allele.

Next, we examined directly whether heterochromatin assembly is impacted. To this end, we focused on the H3K9me2 mark, which signals heterochromatin formation and can accumulate without major changes in gene expression [49]. This allows us to examine whether the loss of nucleation-distal silencing we observe is due to a loss of heterochromatin spreading or a

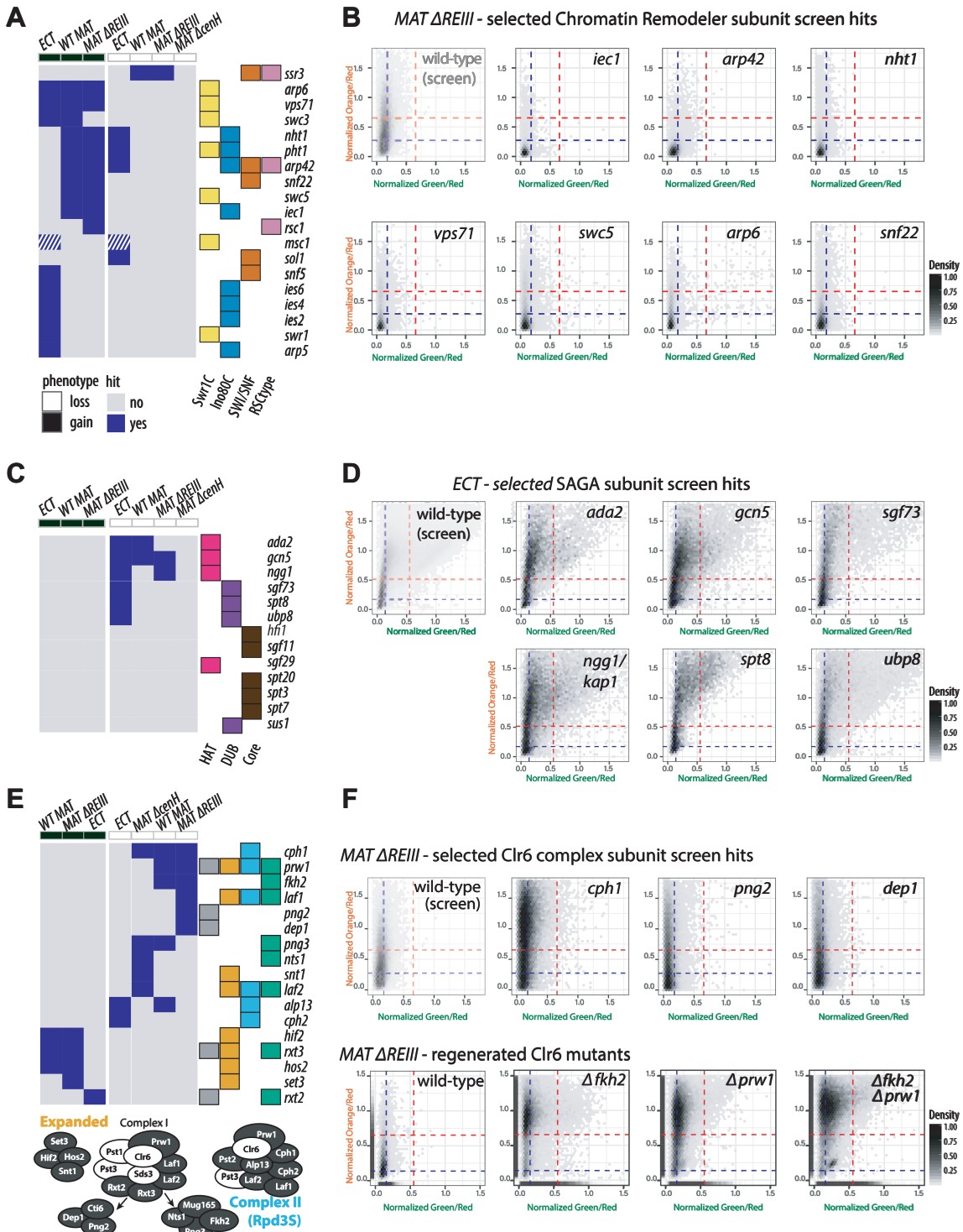

**Fig 4. Chromatin remodeler, SAGA and Clr6 complexes regulate nucleation-distal silencing. A.** Table of complex members which were hits for chromatin remodeler complexes Swr1C, Ino80, SWI/SNF, and RSC. Components that showed a "gain" or "loss" of nucleation-distal silencing phenotype for each background were identified a hits and are marked blue, subunits that are a hit for both phenotypes (*msc1*) are white-blue crosshatched. The proteins present in each complex or subcomplex are annotated at the right: color indicates membership of a particular remodeling complex, as labeled below. **B.** 2D density hexbin plots for selected chromatin remodeler gain of nucleation-distal silencing screen hits in *MAT ΔREIII*. **C.** As described in panel A. but for the SAGA complex. All SAGA subunits

in the screen except TAF$_{II}$s are shown. **D.** 2D density hexbin plots for SAGA loss of nucleation-distal silencing screen hits in *ECT*. **E.** As described in panel A. Clr6 mutants that were not hits are *mug165*, *pst2* and *cti6*. Bottom: schematic of Clr6 complexes with essential subunits indicated in white. **F.** Top: 2D density hexbin plots for selected Clr6 complex loss of spreading screen hits in *MAT ΔREIII*. Bottom: Δ*fkh2*, Δ*prw1 and* Δ*fkh2* Δ*prw1* mutants were re-created *de novo* in *MAT ΔREIII*. A rug plot is included on the X and Y axes indicating the 1D density for each color. Rug lines are colored with partial transparency to assist with visualization of density changes. *MAT ΔREIII* and *ECT* parents shown in Fig 1 are shown here again (with transparency) for comparison.

loss of gene-silencing *per se*. We performed H3K9me2 ChIP-seq analysis in wild-type and the mutants Δ*fkh2*, Δ*prw1*, *clr6-1*, and *clr3-D232N* and analyzed the data in two independent ways: First, we produced input-normalized signal tracks, plotting mean and 95% confidence interval per genotype calculated from multiple replicates (**Fig 5D–5I, top panels**; see Materials and Methods). Second, we conducted a differential enrichment analysis that examines 300bp windows along the genome containing above-background signal for significantly different accumulation of H3K9me2 between each mutant and the wild-type (**Fig 5D–5I, bottom panels**; see Materials and Methods). We define heterochromatin spreading defects as the differential, distance dependent loss of H3K9me2 over genomic features annotated in light grey (non nucleator features) relative to genomic features containing the nucleation elements (annotated in dark grey) (**Figs 5D–5F** and **S9**).

Principal Component Analysis (PCA) on the overall phenotype revealed that mutant isolates segregated from the wild-type along PC1. *Clr6* mutants also diverge from wild-type along PC2 (**S9A Fig**). Focusing on the signal tracks as well as the differential enrichment analysis, we found that the three Clr6-related mutants, Δ*fkh2*, Δ*prw1*, and *clr6-1*, show very strong defects in heterochromatin spreading at most, but not all, heterochromatic locations in the genome (**Figs 5D–5J** and **S9B–S9F**). We independently validated these effects by ChIP-qPCR, additionally examining the Δ*fkh2*Δ*prw1* double mutant (**S10 Fig**). The effects are most prominent at pericentromeres and sub-telomeric regions, as evidenced both by separation of the 95% confidence intervals of the mutant versus wild-type signal tracks and the differential enrichment analysis in the non nucleator regions (light gray, **Figs 5D–5F** and **S9B–S9E**). Clr6 mutants also have strong effects at heterochromatin islands, with severe loss of H3K9me2 (**Fig 5G–5I**). Critically, *Clr6* mutants show minimal defects in H3K9me2 accumulation at nucleation sites (dark gray), especially those driven by RNAi, i.e. *cenH* at MAT, the *dg* and *dh* repeats at the pericentromere, and homologous repeats at the subtelomeric *tlh1/2* gene (**Figs 5D–5F, 5J, S9B–S9E, and S11**), further reinforcing the notion that Clr6 complexes largely do not contribute to heterochromatin assembly during nucleation, but instead are essential for spreading.

Interestingly at the MAT locus, the effect tends to be restricted to regions that are centromere-distal to *cenH* (**Figs 5J** and **S9B**), near the location of "orange" (**S1 Fig**). This effect on spreading does not appear to impact the region near *REII*, which is centromere-proximal relative to *cenH*. This may indicate that Clr6 works redundantly with other regulators at the MAT locus to promote heterochromatin spreading. We also tested if the relatively localized effect of on H3K9me2 to the right side of *cenH* is an artefact of the "orange" reporter insertion. We generated *clr6-1* and Δ*prw1* without any reporter genes and found, similar to *MAT ΔREIII*, a reduction in K3K9me2 accumulation to the right of *cenH* (**S11 Fig**). This indicates that the loss H3K9me2 observed in the ChIP-seq is not due to the insertion of reporters.

In contrast, *clr3-D232N* shows strong H3K9me2 accumulation defects at all major nucleation centers as well as the distal regions (**Fig 3E–3H**), even though some residual H3K9me2 is evident at nucleation centers. This is consistent with the view that spreading is dependent on successful heterochromatin assembly at nucleation sites. The sole exception are heterochromatin islands, where surprisingly, *clr3-D232N* shows increased H3K9me2 accumulation, possibly due to redistribution from RNAi nucleation centers elsewhere (**Fig 5G–5I**).

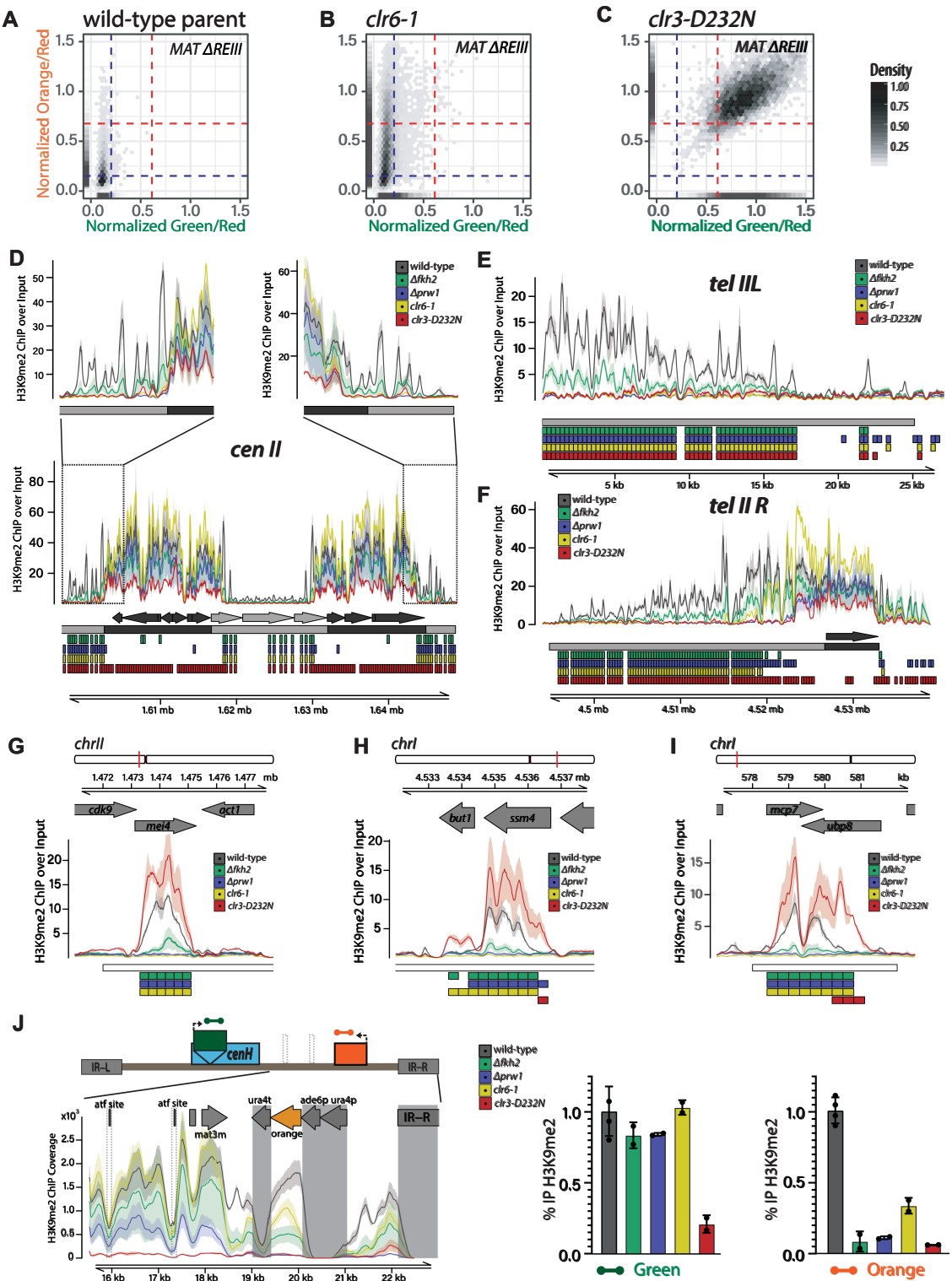

**Fig 5. Fkh2- containing Clr6 Complexes regulate H3K9me2 spreading at constitutive and facultative heterochromatin loci. A.-C.** The hypomorphic *clr6-1* allele exhibits a loss of nucleation-distal silencing, while the catalytic dead *clr3-D232N* allele loses all silencing. 2D density hexbin and rug plots in the *MAT ΔREIII* background of the parent strain (**A.**) run together with B. & C.; *clr6-1* (**B.**); and *clr3-D232N* (**C.**). A rug plot is included on the X and Y axes indicating the 1D density for each color. Rug lines are colored with partial transparency to assist with visualization of density changes. Dashed blue lines indicate the values for repressed fluorescence state and

dashed red lines indicate values for fully expressed fluorescence state. **D.-H.** Visualization of H3K9me2 ChIP-seq signals in the *MAT ΔREIII* background at centromere II (D.), telomere and subtelomere IIL and R (E.&F.), three heterochromatin islands (G.-I.), and the right side of the MAT locus (J. LEFT). ChIP/Input normalized signal is plotted as mean (line) and 95% confidence interval (shade) for each genotype. For J. LEFT, alignment was performed to a custom mating type region contig in *MAT ΔREIII* with green and orange color cassettes. ChIP signal represents the coverage of each interval adjusted for the sequencing depth of the full genome bam file relative to that of the full genome bam file with the lowest depth. Features of interest are annotated above the signal tracks. During data processing for alignment to this custom contig, reads mapping to multiple locations within the reference sequence were removed. For this reason, there is little to no signal over regions that are homologous within this reference including *ura4p/ade6p* at the color cassette promoters, *ura4t* at the color cassette terminators, and *IR-L* and *IR-R* elements (these regions are shaded). Signals at feature "atf site" (indicated by dashed box) are reduced as both 7bp sites in *MAT ΔREIII* are deleted. D.-I. Below the signal tracks the following annotations are present in order from top to bottom: (1) D-F, features of interest (i.e. nucleators, dark grey; non-nucleator, light grey) based on coordinates and strand derived from Pombase (since no nucleator sequences are present on subtelomere IIL, first annotation row below the signal tracks is empty); G-I., previously identified euchromatin embedded H3K9me2 heterochromatin region ("island", "HOOD", or "region") annotated as a white box. (2) for D-F only, nucleation and spreading annotation zones (based on (1)) are represented by dark grey and light grey boxes respectively. Spreading zones are defined to be between or outside of nucleation zones. (3–6, D-F) or (2–5, G-I): 300bp regions determined to be significantly differentially enriched for the comparisons between *Δfkh2* and wild-type (green), *Δprw1* and wild-type (blue), *clr6-1* and wild-type (yellow), *clr3-D232N* and wild-type (red) are annotated as colored boxes respectively. In J. RIGHT, bars above signal tracks indicate wild-type normalized H3K9me2 ChIP-RTqPCR signals for indicated genotypes conducted independently of the ChIP-seq experiment at "green" and "orange" reporters. Error bars represent 1SD of three replicates.

We further analyzed the different impact of Clr6 and Clr3 on H3K9me2 at nucleation sites and distal regions using volcano plots. We compared differential H3K9me2 enrichment in *Δfkh2*, *Δprw1*, *clr6-1*, and *clr3-D232N* relative to wild-type (S12A–S12H Fig). While all mutants have significantly reduced enrichment relative to wild-type at distal sites subject to spreading, these plots reveal key features that separate the phenotype of the Clr6C mutants from *clr3-D232N*: We find, (1) that nucleation center sequences are significantly reduced in the *clr3-D232N* versus wild-type comparison, but not in Clr6 complex mutant compared to wild-type. This further indicates that *clr3-D232N*, but not Clr6 mutants, show significant H3K9me2 accumulation defects at nucleation centers. In addition, this analysis reveals (2) that, in comparison to wild-type, *clr3-D232N* shows significantly increased H3K9me2 enrichment at euchromatic sites, namely islands, which is absent for the Clr6-related mutants (S12D and S12H Fig).

Finally, we assessed if these effects on H3K9me2 in Clr6 mutants are also evident for the major repressive mark, H3K9me3. We conducted H3K9me3 ChIP-qPCR and examined MAT, telomere, islands and a pericentromere II-distal locus (S13 Fig). Our results show that H3K9me3 is depressed at distal sites in *clr6-1* and *Δfkh2* relative to wild-type. This is consistent with our prior work on H3K9me spreading within MAT [24], showing H3K9me3 declines in concert with H3K9me2 at nucleation-distal sites. This is not surprising, since H3K9me3 is required for H3K9me2 spread via the Clr4 chromodomain [18,49]. Overall, these analyses reinforce the view that Clr6 primarily acts by promoting nucleation-distal spreading of H3K9me. This effect is highly localized within MAT, indicating either that Clr6 largely impacts gene silencing functions that are downstream from heterochromatin assembly, or multiple redundant factors work in concert with Clr6 to spread H3K9me at MAT.

## Fkh2 is part of several Clr6 complexes *in vivo*

Fkh2 was previously shown to physically associate with Clr6 [47], and we confirmed that Fkh2 associates with Clr6, and also Sds3 by co-immunoprecipitation (S14A Fig). The co-immuno-precipitation with Sds3 suggests that Fkh2 can associate with Clr6 I complexes, which are typi-fied by Sds3 (S14A Fig) [44]. To assess whether Fkh2 stably integrates into Clr6 complexes, we performed sucrose gradient fractionations of cellular lysates, similar to prior analyses [44], using a Fkh2-TAP fusion and epitope-tagged Clr6 (Clr6-13MYC; Fkh2-TAP) or epitope

tagged Sds3 (TAP-Sds3; Fkh2-13MYC). Our results can be summarized as follows: Fkh2 likely associates with Clr6 complexes Clr6 I′ [44], the related Clr6 I″ complexes [47], as well as Clr6 II. We based this assessment on the fact that Fkh2-TAP co-migrates with at least two Clr6 complexes, a smaller and a larger complex. The large complex migrates in fraction 10 in the Fkh2-TAP: Clr6-13MYC experiment (**Figs 6A** and **S14C**), and peaks in fractions 9–10 in the TAP-Sds3: Fkh2-13MYC experiment (**S14B Fig**). Given that this fraction contains the peak of Sds3, it likely represents a mixture of the large complexes I′ and I″ [44]. Separately, Fkh2 associates with a smaller Clr6 complex in fractions 5–7 in the Fkh2-TAP:Clr6-13MYC experiment (most abundant in fraction 7), which based on prior analysis [44] likely represents Clr6 II. Fkh2-TAP also migrated in an apparent free form (fraction 2, **Fig 6A**), consistent with its role as a transcription factor [50,51]. Next, we sought to address if this co-migration indicates stable Clr6 complex association. We predicted that the migration pattern of Fkh2 would change in mutants of core Clr6 complex members. When we performed sucrose gradient analysis with Fkh2-13MYC (as opposed to Fkh2-TAP above) in the *Δprw1* mutant, we found that the peaks associated with bound, but not free, Fkh2-MYC are shifted towards lower molecular weight by one fraction; this was seen for both, large and small Clr6 complexes (**Fig 6B**). We note that in this experiment, more Fkh2-MYC was detected in lower molecular weight migrating complexes. Therefore, we conclude that Fkh2 is a *bone fide* member of Clr6 complexes.

## Fkh2 helps direct Clr6 to nucleation-distal heterochromatin sites

We next sought to understand how Fkh2 helps Clr6 spread H3K9me2 or gene silencing. One trivial possibility is that Fkh2 promotes the transcription of major heterochromatin components, independent of its association with Clr6 complexes. To test this notion, we performed RT-qPCR for 9 heterochromatin regulators, which were chosen to represent the major complexes ClrC, RITS, Clr6, and SHREC. We also separately queried the key heterochromatin assembly factor gene *swi6*. We do not observe any significant reduction of these transcripts in *Δfkh2* compared to wild-type (**S15 Fig**), suggesting that Fkh2 acts via another mechanism. We next tested whether Fkh2 affects the chromatin localization of Clr6. Using ChIP, we tracked the chromatin association of Clr6-13MYC at various heterochromatic loci in WT or *Δfkh2* in the *MAT ΔREIII* heterochromatin spreading sensor background. At the MAT locus, Clr6-13MYC was efficiently detected above background signal (untagged control) at *cenH* "green" or at *mtd1*, a euchromatic gene outside the *IR-R MAT* boundary. At these loci, Clr6 recruitment was unaffected by the absence of Fkh2 (**Fig 6C**). However, at "green"-distal sites, namely the "orange" reporter and a more boundary proximal site ("MAT distal"), *fhk2* deletion reduced Clr6 localization. Similarly, *Δfkh2* affected Clr6 localization at the heterochromatin islands *mei4*, *mcp7*, and *ssm4* (**Fig 6D**). At telomeres, where *Δfkh2* also had a significant impact on H3K9me2 spreading, we also found a significant reduction in Clr6 chromatin association at distal sites (**Fig 6E**, 30kb) but not at the euchromatic control locus, *act1* (**Fig 6F**). Therefore, it appears that one role of Fkh2 in promoting heterochromatin spreading is to strengthen the recruitment of Clr6 to nucleation-distal heterochromatic sites.

## Discussion

The formation of a heterochromatin domain requires three interconnected steps: DNA-sequence driven nucleation, assembly of heterochromatin structures, and the lateral spreading to neighboring regions. It remains poorly understood whether the distal propagation of gene silencing or the heterochromatin structure itself has locus-specific requirements, and whether the genetic circuitry directing proximal and distal events overlap or are separable. Our reporter allows us to separate requirements for nucleation-proximal and distal silencing and thereby

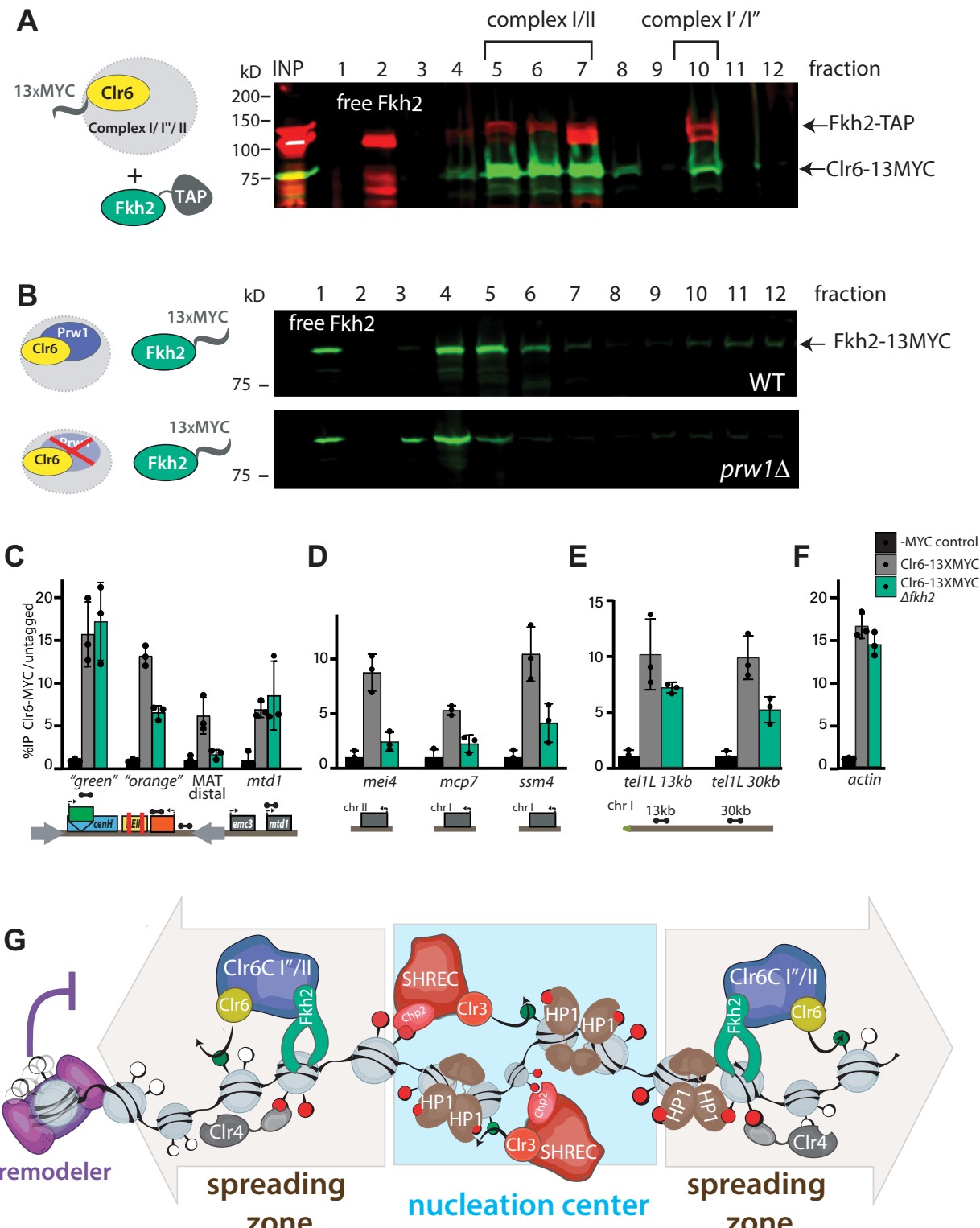

**Fig 6. Fkh2 is a resident member of multiple Clr6 complexes and directs Clr6 to nucleation-distal heterochromatin regions. A.** Fkh2 co-migrates with medium sized and large Clr6 complexes. Western blot against TAP (red) and MYC (green) on fractions of a sucrose density gradient of whole cells extract containing Fkh2-TAP and Clr6-MYC. Fkh2 migrates as a free protein on top of the gradient and in both major peaks of Clr6. Complex annotation based on [44]. **B.** Fkh2 migration in sucrose gradients depends on Prw1. Western blot against MYC (green) on fractions of a sucrose density gradient of wild-type (WT) or Δ*prw1* whole cell extract containing Fkh2:MYC. While the free Fkh2 fraction is

not affected by the absence of Prw1, the medium and large peak fractions shift towards the top of the gradient. **C.-F.** Clr6 chromatin localization in heterochromatin spreading areas and heterochromatin islands depends partially on Fkh2. Clr6:MYC ChIP qPCR in wild-type (Clr6-13XMYC) or Δ*fkh2* (Clr6-13XMYC Δ*fkh2*) background relative to an untagged (-MYC) control, in *MAT ΔREIII* at the HSS reporter and downstream (**C.**), heterochromatin islands (**D.**), tel1L (**E.**), and *act1* (**F.**). Error bars indicate 1SD of three replicates. **G.** Model for control of heterochromatin spreading by Clr3, Clr6 and chromatin remodeler complexes. The Clr3- containing SHREC complex is essential for heterochromatin assembly at noncoding RNA-driven nucleation centers, such as *cenH*, *dg/dh* repeats or *tlh1/2*. There, SHREC activity enables normal HP1 and Clr4 H3K9 methylase activity. In the spreading zone, Fkh2-containing Clr6 complexes (complex I″ or complex II) are required to propagate H3K9 methylation and gene silencing. Fkh2 recruits Clr6 to nucleation center-distal chromatin. Chromatin remodelers repel spreading via nucleosome destabilization, thus hinder the "guided-state" nucleosome-to-nucleosome spreading mechanism of Clr4.

pinpoint which factors are necessary to drive or restrain it at different genomic loci. A key finding is that variants of the Clr6 HDAC complex are specifically involved in distal heterochromatin spreading and/or silencing. In contrast, a broad class of chromatin remodelers antagonize nucleation-distal heterochromatin silencing. Additionally, we find that different chromatin contexts have specific requirements for distal silencing. For example, the SAGA complex promotes heterochromatin spreading in a euchromatic context.

Several HDACs are involved in heterochromatin regulation in fission yeast, Clr6, Clr3 and Sir2. Of those, we find that Clr6 has a disproportionate effect on nucleation-distal sites. In contrast, the striking loss of silencing and H3K9me2 accumulation at nucleation sites in *clr3* mutants indicate that Clr3 either primarily impacts nucleation, which precedes spreading, or that it affects both processes (**Figs 1E, 1G, 1I, and 5** and **S9**). We note that while Clr3 is thought to be required for silencing at nucleation centers, for example at *REIII* and *cenH* [2,26,45], the literature on the impact of *clr3* gene deletions on the distribution of H3K9me2 is mixed, showing either no [2] or significant loss [31]. We believe that the strong loss of H3K9me2 we observe in *clr3-D232N* closely reflects the native function of the gene, given that prior analysis of this mutation indicates a very similar degree of loss of function [45]. The variable results for *Δclr3* may reflect compensation that occurs in strains carrying null versus other loss-of-function alleles [52]. Our finding that Clr6 only affects distal silencing is consistent with the finding that the *clr6-1* allele has only small impacts on transcription of the *cenH* nucleator-encoded ncRNAs [53]. Our screen showed that several, but not all members, of the recently described Clr6 complex I″ [47] and, to a lesser extent, other Clr6 complexes, promote spreading. Not all annotated Clr6 subunits share gain or loss of spreading phenotypes, suggesting that these subunits do not contribute to distal silencing but instead mediate other functions of the complex.

Fkh2 prominently promotes spreading of H3K9me2 at pericentromeres and sub-telomeres and at the right side of the *MAT* locus. We find that Fkh2 is a resident member of several Clr6 complexes, the I′,I″, and II types (**Fig 6A** and **6B**). These results indicate two possible, nonexclusive interpretations: 1. the composition of different Clr6 subcomplexes *in vivo* is more dynamic than previously thought; 2. a number of different, preassembled Clr6 complexes can associate with Fkh2, which imparts a role in spreading regulation. Interestingly, the Set3-submodule that typifies the Rpd3L-Expanded complex [54] has a distinct spreading-antagonizing behavior (**Fig 4C**). This is in contrast with prior findings on the Set3 complex, where it was shown to exhibit a mild positive role at pericentromeres [55]. Taken together, our data suggest a division of labor between Clr3 and Clr6, with Clr3 majorly impacting heterochromatin assembly at nucleation sites and Fkh2-containing Clr6 complexes function to spread heterochromatin structures and/or silencing outward (**Fig 6G**). However, a formal possibility remains that these deacetylases function in similar ways, with the activity of Clr6 at nucleation sites masked by the activity of other HDACs. The observation that heterochromatin spreading at the MAT locus is weakly affected while gene silencing is strongly affected indicates that other pathways, such as FACT [38,56], may act redundantly with Clr6 in heterochromatin

spreading. It is also possible that Clr6 is primarily involved in distal propagation of gene silencing via its deacetylation function, which when lost indirectly weakens heterochromatin at certain loci.

What mediates the spreading-specific role of Clr6 complexes? One possible explanation is that they act in conjunction with the histone chaperone Asf/HIRA, which cooperates with Clr6 in gene silencing at ncRNA nucleators [53]. However, we do not favor the idea that this pathway mediates distal silencing, since Asf/HIRA subunits Hip1, Hip3 and Slm9 have mild or no phenotypes for distal silencing in *MAT* contexts. Asf/HIRA mutant phenotypes were more pronounced in *ECT*, a context that is less reliant on Clr6 for spreading (**Fig 1,** see below). We believe that Fkh2 plays a key role in imparting this spreading-specific role for Clr6. Fkh2 is a transcription factor that regulates meiotic genes in *S. pombe* [50,51,57]. However, the observation that loss of Fkh2 impacts H3K9me2 spreading at non-nucleator sites (over grey feature annotations, **Figs 5 and S9B–S9E**) raises two possibilities: 1. Fkh2 acts via its canonical sequence binding capacity. In this case, Fkh2 may exhibit DNA binding that is more degenerate than the consensus motif, or it may bind its canonical consensus sequence and regulate chromatin architecture. In support of the first, while Fkh2 is described as a DNA binding transcription factor that recognizes specific motifs [57,58], Fkh2 appears to be found on many regions of the chromosome, including at the left side of the pericentromere (*cen II*), the MAT locus and several other locations (genome browser, pombase tracks from [58]). Alternatively, rather than binding locally along chromatin, Fkh2 may also act as a chromatin organizing protein, as has been previously shown [59]. It is conceivable that Fkh2 creates chromatin environments that are conducive to nucleation-distal Clr6 recruitment, either by tethering to a nuclear compartment such as the nuclear periphery [38] or via chromatin conformational or biophysical changes [60,61]. 2. It is possible that Fkh2 acts in a transcription factor-independent fashion, recognizing features of nucleation-distal regions through regions other than its sequence-specific binding domain. Instances of transcription factors executing key functions independent of DNA binding, but via protein-protein interactions, have been demonstrated for example in plants [62]. We find that Fkh2 promotes the recruitment of Clr6 complexes to the spreading zone (**Fig 6C–6E**). The spreading-specific role of Clr6 complexes may be additionally supported by other known recruitment mechanisms, for instance a pathway involving HP1/Swi6 [1,5,63]. While the precise mechanism of Fkh2-mediated Clr6 recruitment will be the subject of further studies, our results unambiguously demonstrate that Fkh2 typifies a specific functional mode of the conserved HDAC Clr6/RPD3 among its numerous essential activities in gene regulation [64]. Given that Fkh2 resides within Clr6 complexes (**Fig 6A** and **6B**), we favor the view that it directly participates in regulating spreading of gene silencing or H3K9 methylation.

Beside the positive regulation of heterochromatin spreading detailed above, our results show that several classes of chromatin remodelers, including Ino80, Swr1C, SWI/SNF and RSC, antagonize nucleation-distal silencing. This broad antagonism contrasts with more specific functions uncovered previously for Ino80/Swir1C [37]. As seen for Clr6, not all subunits in remodeling complexes show a spreading phenotype. Remodelers have been implicated in negatively regulating heterochromatin function by creating specific nucleosome free regions (NFRs, [65]) that antagonize heterochromatin. Since NFRs may be roadblocks to spreading [66,67], it is possible that remodelers employ this mechanism to restrain heterochromatin spreading globally. In addition, remodelers such as SWI/SNF and RSC generally destabilize nucleosomes [68,69], leading to increased turnover [70], which would antagonize heterochromatin spreading. This increased turnover may be tolerated at ncRNA nucleation sites, where turnover is at near euchromatic levels [24], likely due to ncRNA transcription [29,71]. This would suggest that regulation of nucleosome stability has a particular significance at distal, but not nucleation sites.

Beyond the finding of Clr6 and remodelers in promoting and antagonizing heterochromatin spreading, respectively, this study uncovered several locus- and nucleator-type-specific pathways. Here we would like to highlight two main observations:

1. Distinct factors are required for similar nucleators in different chromatin environments. *ECT* and *MAT ΔREIII* are both driven by related ncRNA nucleators (*dh* and *cenH*, respectively) and have remarkably similar behaviors with respect to nucleation and spreading across the cell population [24]. Efficient spreading, specifically at *ECT*, requires Hip1, and moderately Slm9, which code for a key subunits of the HIRA H3/H4 chaperone. HIRA has been implicated in stabilizing heterochromatic nucleosomes [53]. Hence, given that transcribed chromatin is known to destabilize nucleosomes, it seems likely that this specific requirement reflects the challenge faced by heterochromatic domains when expanding within gene-rich chromatin. *ECT* is also particularly reliant on the SAGA complex for spreading (**Figs 4A** and **S4**). This may seem counterintuitive initially, as SAGA has been shown to be recruited by Epe1 to antagonize heterochromatin assembly at constitutive sites [72]. In fact, in our screen SAGA plays a less prominent role in the MAT context. This requirement for SAGA at *ECT* may be connected to the observation that SAGA can modulate the chromatin recruitment of remodelers, such as SWI/SNF, via direct acetylation [73]. Therefore, one possible explanation that remains to be tested for the SAGA phenotype we observe is acetylation of SWI/SNF and possibly other remodelers that antagonize spreading, releasing them from chromatin.

2. Spreading from qualitatively different nucleators within the same environment, namely *REIII* and *cenH*, also differs in sensitivity to different mutants. The significant overlap in factors between *WT MAT* and *MAT ΔREIII* indicates that heterochromatin formation at MAT is dominated by the ncRNA nucleator *cenH*, in agreement with our previous findings [24]. The *REIII* element, which nucleates heterochromatin independent of ncRNA [23], has different requirements. For example, ncRNA-independent spreading at *REIII* (MAT *ΔcenH*) is uniquely promoted by the TORC2 pathway Gad8 kinase, consistent with a previous report implicating Gad8 for MAT silencing [74]. While Gad8 is reported to target Fkh2 for phosphorylation, *Δfkh2* has very weak effects on MAT *ΔcenH*. Other potential phosphorylation targets of Gad8 in promoting spreading from *REIII*, if it acts through its kinase function, remain to be established. We note that *REIII* can confer a high propensity for local intergenerational inheritance of silencing [24,39]. Therefore, a formal possibility for spreading defects in the MAT *ΔcenH* context, or others with high intergenerational stability, is that the genes identified are required for the maintenance of silencing outside the nucleator, rather than the initiation of silencing across the distal locus.

In this work, we defined the regulation of nucleation-distal gene silencing, which has specific chromatin context-dependent requirements that are separate from the regulation of silencing at nucleation centers. While similar nucleation elements likely rely on common mechanisms, the success of heterochromatin mediated distal silencing appears to depend on the chromatin context and particularly differs in gene-rich versus gene-poor chromatin. Our findings have implications for directing gene silencing during cellular differentiation. In this situation, regions that have previously been in a transcriptionally active state are invaded by heterochromatin and will therefore compete for core spreading factors in a dosage limited system [75,76]. We note that several of the factors we identify as critical to regulating spreading in euchromatinic environments are conserved in metazoans, indicating that they may contribute to differentiation through heterochromatin control in these organisms.

## Materials and methods

### Mutant generation for genetic screen

We generated a 408 gene deletion mini-library that represents a subset of the *S. pombe* Bioneer deletion library, focused on nuclear function genes. The library (**S1 Table**) was assembled via

three criteria: 1. ~200 genes coding for proteins have a "nuclear dot" appearance in a high-throughput YFP- tagged "ORFeome" screen ([77], also used in [34]). 2. ~50 deletions of central *S. pombe* DNA binding transcription factors. 3. ~150 gene deletions selected based on their annotation as chromatin regulation-related functions (majority of this set), or prior preliminary data indicating a role in heterochromatin function. Deletions that grow poorly in rich media were eliminated. Several Bioneer collection mutants were independently validated and produced *de novo*. For the ectopic locus HSS reporter strain, the screen was performed essentially as described [8]. Briefly, the parent HSS reporter strain was crossed to the library. Crosses were performed as described [8,24,40,78] using a RoToR HDA colony pinning robot (Singer). For the MAT HSS reporter strains, the screen was performed essentially as described [8] with the exception that crosses were generated using a 96 well manual pinner. Note that the *MAT* we used an OFF isolate for *ΔcenH* [24]. This is because *ΔcenH* behaves in a bimodal fashion, producing stable ON (no heterochromatin) and OFF (heterochromatin) alleles at *MAT*. *MAT ΔcenH* isolates were picked absent any selection based on their "green" and "orange" profiles in flow cytometry [24].

In addition, three *Δclr4* mutant isolates and six individual parent isolates from each genomic context were included as controls. Crosses for the ectopic HSS strains were performed using SPAS media for 4d at room temperature, while for the MAT HSS strain crosses were performed on ME media for 3d at 27°C. For all strains, crosses were incubated for 5d at 42°C to retain spores, while removing unmated haploid and diploid cells. The ectopic HSS spores were germinated on YES medium supplemented with G418, hygromycin B, and nourseothricin. For MAT HSS strains, spores were germinated on YES medium supplemented with G418 and hygromycin B. The resulting colonies were pinned into YES liquid medium for overnight growth and then prepared for flow cytometry as described below.

## Flow cytometry data collection and normalization for genetic screen

In preparation for flow cytometry, overnight cultures were diluted to OD = 0.1 (approximately a 1:40 dilution) in rich media (YES) and incubated at 32°C with shaking of rpm for 4–6 hours. For the ectopic locus HSS strains, flow cytometry was performed essentially as described [8,24]. For the MAT locus HSS strains, flow cytometry was performed using a Fortessa X20 Dual instrument (Becton Dickinson) attached with high throughput sampler (HTS) module. With a threshold of 30,000 events, samples sizes ranged from ~1000 to 30,000 cells depending on strain growth. Fluorescence detection, compensation, and data analysis were as previously described [8,24]. Flow data derived from the genetic screen for individual strains are represented as 2d density hexbin plots in Figs 1, 2, 4, and 5 and S2, S4, S7, and S8 Figs. Dashed red and blue guidelines respectively indicate median minus 2SD of "green"/"orange"-On cells (*Δclr4* control cells with normalized fluorescence above 0.5) and median plus 2SD of "green"/ "orange"-Off cells ("red"-Only control).

## Spreading analysis

Nucleated cells were extracted using a "green"-off gate, using median of a "red"-only control plus 2 times the SD. Enrichment of cell populations in particular "orange" fluorescence ranges ($Grid_n$) are calculated as $Grid_n^{mut/par}$: fraction of mutant population is divided by the fraction of parent population in one grid. The intervals of "orange" fluorescence used in grids are determined by: median plus 2SD of "orange"-Off cells ("red"-Only control), median minus 2 SD of "orange"-On cells (*Δclr4* control cells with normalized fluorescence above 0.5) and the median of the two. For *ECT*, instead of using a "red"-Only control strain, we adjusted the fluorescence value of a fluorescence-negative (unstained) strain by the "red" fluorescence from the *Δclr4*

control. To evaluate gain of spreading phenotype, enrichment in Grid 1 in *WT MAT*, *MAT ΔREIII* and *ECT* were calculated. To evaluate loss of spreading phenotype, enrichment in Grid 3 and 4 in *MAT ΔcenH* and *WT MAT* as well as Grid 4 in *MAT ΔREIII and ECT* were calculated. The distribution of the $Grid_n^{mut/par}$ were plotted as swarmplots with annotation of the 85th percentile and median plus 2SD of parent isolates $Grid_n^{mut/par}$. Gene hit lists comprised mutations above median and 2SD within the 85th percentile. Upset plots were generated using the R package UpSetR [79]. Beeswarm plots were plotted using the R packages ggbeeswarm [80].

## GO complex and sub-complex analysis

**Generating the heatmap count data.**    GO Complexes–Based on the GO Complex annotations [link] retrieved from pombase [42,81], GO complex membership was determined for genes identified as hits for each strain background and hit category (gain/loss). Briefly, using functions from the R package dplyr (Wickham H., François R., Henry L. and Müller K. (2020). dplyr: A Grammar of Data Manipulation. R package version 0.8.4. https://CRAN.R-project.org/package=dplyr), gene names were converted to systematic ID numbers and these systematic IDs were queried against the GO complex annotation table. The number of times a GO complex appeared per background and hit category was tabulated. Genes can be associated with any number of GO complexes depending on their annotations. However, each gene was only counted once per GO complex despite potentially being annotated to that GO complex by more than one evidence code. The unique list of GO complexes for all hits was determined and a matrix was computed representing the number of times each GO complex (row) was identified per strain/hit category (column). This counts matrix was used to generate the GO complex heatmap in Fig 3, described below.

Hit tables–Genes annotated to the seven complexes in Fig 4A and 4C and 4E were obtained from pombase [42]. *fkh2* was added to the Clr6 I″ complex given the protein contacts described previously [47]. For the unique set of genes per panel it was determined if each gene was identified as a hit in each strain background/hit category combination. The data was summarized in a counts matrix where rows represent the unique list of genes per panel and columns represent the strain background / hit category. The counts matrix for each set of genes was used to generate the heatmaps in Fig 4A and 4C and 4E as described below.

**Generating the heatmap clustering.**    Using the R package ComplexHeatmap [82], both row and column dendrogram and clustering were generated using hierarchical clustering. Based on an optimal Silhouette score, the strain background / hit category (columns) were clustered into 2 clusters. The dendrogram representing complexes (Fig 3) or genes (Fig 4A and 4C and 4E) in rows were not separated because validations of the clustering by connectivity, Dunn index or Silhouette score were inconclusive. Clustering validations were conducted using the R package clValid (Brock, G., Pihur, V., Datta, S. and Datta, S. (2008) clValid: An R Package for Cluster Validation Journal of Statistical Software 25(4) URL: http://www.jstatsoft.org/v25/i04).

## Validation of strains and plasmid construction

Plasmid constructs for gene knockout validation in HSS background strains were generated by *in vivo* recombination as described [8,24]. *S. pombe* transformants were selected as described [24]. For microscopy, *hygMX* super-folder GFP (SFGFP) constructs for C-terminal tagging we described previously [33] were amplified with 175bp ultramer primers with homology to *apm3* or *apl5* and transformed into a Swi6:E2C *kanMX* strain. Apm3:SFGFP;Swi6:E2C and Apl5:SFGFP;Swi6:E2C strains were selected on hygromycin B and G418. For validations in

PAS100 (wild-type, no HSS): *clr6-1* (PAS933) was generated by CRIPSR/Cas9 editing as described [83]. *Δprw1* (PAS932) was generated by first outcrossing the HSS from PAS799 and then crossing the progeny to PAS100. Integrations and gene knockout were confirmed by PCR and sequencing (*clr6-1*). Strains generated for this study beyond the screen can be found in S2 Table.

## Flow cytometry data collection and normalization for validation

For validation of flow cytometry experiments, cells were grown as described [8,24] with the exception that cells were diluted into YES medium and grown 5–8 hours before measurement. Flow cytometry was performed as above. Depending on strain growth and the volume collected per experiment, fluorescence values were measured for ~20,000–100,000 cells per replicate. Fluorescence detection, compensation, and data analysis were as described [8,24,33] with the exception that the guide-lines for boundary values of "off" and "on" states were determined using median of a Red-Only control plus 3 times the median absolute deviation (MAD) and median of *Δclr4* minus 2 times the MAD value respectively. Validation flow cytometry plots were generated using the ggplot2 R package [84].

## Chromatin immunoprecipitation and quantification

Chromatin Immunoprecipitation (ChIP) was performed essentially as described [8,24]. Bulk populations of cells were grown overnight to saturation in YES medium. For anti-H3K9me2 ChIP, the following morning, cultures were diluted to OD 0.1 in 25mL YES and grown for 8h at 32˚C and 225rpm. Based on OD measurements, $60x10^6$ cells were fixed and processed for ChIP as previously described [24] without the addition of W303 carrier. For anti-MYC ChIP and anti-H3K9me3 ChIP, 40mL cultures were grown to OD 0.4–0.7 and then incubated Cells for anti-MYC ChIP were lysed as described [24], except that cells were bead mill homogenized for 9 cycles. Cleared chromatin for anti-H3K9me2, anti H3K9me3, or anti-MYC ChIP samples was incubated with either 1μL of anti-H3K9me2 antibody (Abcam, ab1220), 1μL of anti-H3K9me3 antibody (Millipore 07–442, lot 3782120), or 2μL anti-MYC antibody (Invitrogen, MA1-980, lot VL317116) overnight after a small fraction was retained as Input/WCE. DNAs were quantified by qPCR. For H3K9me2 and H3K9me3, percent immunoprecipitation (%IP, ChIP DNA/Input DNA) was calculated as described [24], except for S6 Fig, where a ratio of % IP queried locus/%IP *act1* is plotted. For anti-MYC ChIP, enrichment is presented as the ratio of %IP in PAS867 or 868 (Clr6:13XMYC in PAS 332 WT or *Δfkh2*) over %IP in PAS332 (untagged). Data was plotted in Prism (GraphPad). For comparison of different preparations of ChIP samples, %IP of mutant divided by %IP of wildtype was calculated.

## ChIP-seq data collection, library preparation and sequencing

ChIP was performed essentially as above, with the following exceptions: From 60 mL cultures, $300x10^6$ cells in logarithmic phase were fixed and processed. Sheared chromatin samples were not pre-cleared with Protein A Dynabeads, and the chromatin was directly treated with 2μL of anti-H3K9me2 antibody (Abcam 1220, Lot GR3308902-4). Barcode-indexed sequencing libraries were generated from reverse-crosslinked ChIP-DNA samples using a Kapa Hyper DNA Library Preparation Kit (Kapa Biosystems-Roche, Basel, Switzerland) and NextFlex UDI adapters (PerkinElmer, Waltham, MA). The libraries were amplified with 16 PCR cycles and cleaned with SPRI bead protocol according to the instructions of the manufactures. The fragment lengths of the sequencing libraries were verified via micro-capillary gel electrophoresis on a LabChip GX Touch system (PerkinElmer). The libraries were quantified by fluorometry on a Qubit instrument (LifeTechnologies, Carlsbad, CA), and combined in a pool at equimolar

ratios. The library pool was size-selected for library molecules in the lengths of 200 to 450bp using a Pippin-HT instrument (Sage Science, Beverly, Massachusetts). The success of the size-selection was verified on a Bioanalyzer 2100 instrument (Agilent, Santa Clara, CA). The pool was quantified with a Kapa Library Quant kit (Kapa Biosystems-Roche) on a QuantStudio 5 real-time PCR system (Applied Biosystems, Foster City, CA) and sequenced on a Illumina NextSeq 500 (Illumina, San Diego, CA) run with paired-end 40bp reads.

### ChIP-seq data analysis

Data processing for ChIP-seq analysis was performed as follows. Trimming of sequencing adaptors and sliding window quality filtering were performed using Trimmomatic v0.39 [85]. Filtered and trimmed paired-end (PE) reads were aligned to the *S. pombe* genome (Wood et al. 2002) with Bowtie2 v2.4.2 [86] using standard end-to-end sensitive alignment. An additional 6bp was trimmed from the 5' end of each read prior to alignment. Sorted, indexed bam files were generated using SAMtools v1.12 [87]. Duplicate reads were marked with Picard tools v2.25.2 "MarkDuplicates" command. Filtered bam files were generated with SAMtools "view" with the following flags [-bh -F 3844 -f 3 -@ 4 I II III mating_type_region] to retain only properly paired reads on the listed chromosomes/contigs, and remove duplicate reads. The resulting filtered bam files were sorted and indexed with SAMtools and used for downstream genome-wide analysis. Input-normalized BigWig files for signal tracks for 25bp bins were generated from the filtered bam files with the bamCompare function from deeptools v3.5.1 [88]. To do so, the following flags were used: [—outFileFormat bigwig—scaleFactorsMethod readCount—operation ratio—pseudocount 1—extendReads—samFlagInclude 64—skipZeroOverZero—binSize 25—numberOfProcessors 4—effectiveGenomeSize 12591546 –exactScaling]. Fragments were counted once by including only the first mate of each pair and extending to the fragment size. For each genotype, 2 or 3 biological replicates were processed for downstream analysis. Initial whole genome clustering analyses on ChIP and Input samples and inspection for low signal to noise ratio in IGV prompted us to remove two outlier samples, one *clr6-1* and wild-type respectively.

BigWig files were imported into R v4.0.3 with rtracklayer v1.50.0 [89]. BigWig files were used to generate signal tracks comprised of the mean and confidence interval for each genotype in custom genome browser plots generated with the DataTrack() command from the Gviz Bioconductor package v1.34.1 [90]. As described previously (8), gene annotations were imported from PomBase [42] and converted to genomic coordinates in R v3.5.1 with the make-TxDbFromGFF function from GenomicFeatures v1.32.3 [91] and saved out to an sqlite file. This sqlite file was imported into R v4.0.3 and used to generate feature annotations for signal tracks in Gviz with the AnnotationTrack() command.

Reads from the filtered bam files for ChIP samples were counted into either 300bp windows or 5kb bins using windowCounts() function from the Bioconductor package csaw v1.24.3 [92]. Regions +/-1.5kb surrounding the following features (*ade6*, *ura4*, *fkh2*, *prw1*, *mat1-Mc*, *mat1-Mi*) were blacklisted as they represent experimental artifacts–*ade6* and *ura4* because their promoter and terminator regions are present in the reporter cassettes, *fkh2* and *prw1* because these gene's ORFs were entirely removed in certain genetic backgrounds and mat1 genes because these regions are expressed but homologous to those found in the heterochromatic MAT locus. Global background was determined from 5kb bin count matrices and interpreted onto the 300bp windows with the csaw filterWindowsGlobal() function. Only 300bp windows where the abundance exceeded a filtering threshold of 1.7 times the global median were retained resulting in 1500 windows for further analysis.

Differential Enrichment analysis was performed on these 1500 300bp windows using the DESeq2 Bioconductor package v1.30.1 [93]. Size Factors were calculated on the count matrices in the global 5kb bins with the estimateSizeFactors() function and applied on the global enrichment filtered 300bp windows. The model matrix for the experimental design was constructed based on genotype. Normalized counts per bin were obtained with the vst() function from DESeq2 with the parameter blind = FALSE. VST transformed counts were used as an input to principal component analysis via the prcomp() R function on the top 500 most variable bins (as adapted from RNA-Seq differential enrichment workflows), or all 1500 bins. PCA plots were generated with ggplot2 v3.3.3 [84]. Differential Enrichment analysis, including estimating dispersions and fitting of a negative binomial generalized linear model, was performed with the DESeq() function. Results for pairwise contrasts between genotypes were extracted as GRanges objects with the results() function. Each of the 1500 windows was annotated as belonging to global or heterochromatin location specific nucleation/spreading/euchromatic/ other categories. Volcano plots were generated with ggplot2 for each comparison by plotting -log10(padj) against log2FoldChange values. Each dot represents a 300bp region tested for differential enrichment. Dots are colored by their location annotation. Coordinates for nucleator regions are derived from feature coordinates from PomBase adjusted to a multiple of 300 so that a 300bp bin can only be annotated to one category of feature. Coordinates for spreading are defined to be between or outside of nucleator regions. Euchromatic regions are defined as coordinate ranges identified as an "island" [20], "HOOD" [94], or "region" [95] as delineated in *Supplemental Table S6* from [96]. A threshold for significance of padj (Benjamini-Hochberg adjusted p-value) $< 0.005$ and abs(log2FoldChange) $>$ log2(2) was applied to the results for each comparison and these cutoff values are additionally annotated on the volcano plots. For each comparison, the number of significant regions of each category was tabulated. Regions called as significant for the comparison of each mutant to WT are annotated on the custom genome browser plots generated with the Gviz AnnotationTrack() command. A Venn diagram of regions where WT signal significantly exceeds signal from mutants was generated outside of the conda environment in R v4.0.5 with the venn R package v1.10.

An additional custom analysis for the MAT locus was performed starting at the genome alignment stage. A fasta file that includes the inserted "green" and "orange" color cassettes and intact *atf1/pcr1* binding sites was used was used to build a genome index for bowtie2. Alignment to this custom reference was performed with bowtie2 as above. Alignments were filtered with SAMtools to retain only properly paired reads. Multimapping and low-quality alignments were removed with a mapq filter of -q 10. In the context of this custom reference sequence, multimapping reads represent regions that align to *ura4p*, *ade6p*, and *ura4t* which are present at XFP reporter cassettes, *IR-R* and *IR-L* repetitive regions, and parts of the mating type cassettes. Duplicate reads were marked and removed with Picard tools. Sorted, and indexed bam files were generated with SAMtools. BigWig files for coverage signal tracks at 10bp resolution were generated from ChIP files using the deeptools bamCoverage function with the following flags [—outFileFormat bigwig—scaleFactor ##—extendReads—samFlagInclude 64—binSize 10—numberOfProcessors 4—blackListFileName cenH_blacklist.bed—effectiveGenomeSize 19996—exactScaling]. Bam files were scaled by a custom scaling factor (replacing ## above) that adjusts for the read count in the bam file from each sample's the full genome alignment to the read count of the bam file with the smallest number of reads in the full genome alignment. The regions included in the *cenH* element were blacklisted in this step as they are homologous to many sequences found at the centromeres and telomeres and in this analysis represent aggregate signal from all these regions. The resulting coverage BigWig files were used to generate signal tracks in R/Gviz as described above.

## Microscopy

Swi6:E2C; Apl5:SFGFP and Swi6:E2C; Apm3:SFGFP cells were grown is YES media as described. Slides (ibidi, Cat. No. 80606) were pre-coated with 100 mg/mL lectin (Sigma-Aldrich, Cat. No. L1395) diluted in water by adding lectin solution to slide for 1 min. and removing supernatant. Cells growing in log-phase were applied to the slide and excess cells were rinsed off with YES. Cells were immediately imaged with a 60x objective (CFI Plan Apochromat VC 60XC WI) on a Nikon TI-E equipped with a spinning-disk confocal head (CSU10, Yokogawa) and an EM-CCD camera (Hammamatsu). Cells were imaged in brightfield and also excited with 488nm (SFGFP) and 561nm (E2C) lasers. Emission was collected using a 510/50 band-pass filter for GFP emission and a 600/50 band-pass filter for E2C emission. For the SFGFP and E2C channels, z-stacks were obtained at 0.3μm/slice for 11 slices total. An overlay of the maximum z-projections for SFGFP and E2C channels are shown separately from the brightfield images. Brightness and contrast were adjusted in ImageJ to clearly show both Swi6 and Apl5/Apm5 signals in the overlay. At least two isolates were imaged to confirm localization patterns.

## Reverse transcription qPCR validation of context-specific spreading mutants

For validation of context-specific spreading hits, *saf5*, *eaf6*, *pht1*, *hip1* and *gad8* mutants were crossed to PAS217 (WT-MAT), PAS332 (MAT-ΔREIII), PAS482 (MAT-ΔcenH), and PAS231 (*ECT*), respectively and mating products selected for KAN and HYG (PAS217, PAS332, and PAS482) or KAN, HYG, and NAT resistance (PAS231). For analyzing transcriptional regulation of heterochromatin regulators via Fkh2, PAS332 or PAS798 (Δfkh2 in PAS332) were grown as above. Two independent isolates of each strain were grown in 200μl YES in 96 well plates to log phase (OD~0.4–0.8), washed with water and flash frozen. Total RNA was extracted from cell pellets as described [24] using the Masterpure yeast RNA extraction kit (Lucigen). cDNA was produced from 2–3μg total RNA as described [24] using a dT primer and Superscript IV (Invitrogen) reverse transcriptase, followed by an RNaseH step to remove RNA:DNA hybrids. qPCR was performed with primers against *act1*, SF-GFP and mKO2, or amplicons for heterochromatin regulator transcripts indicated in S15 Fig. For *act1* qPCR, the cDNA was diluted 1:60. qPCR was performed as described [8]. qPCR amplicon primers can be found in S3 Table.

## Sucrose gradient analysis

Sucrose gradient analysis was performed essentially as in [97], with several modifications. PAS833 (Fkh2:13XMYC), PAS836 (Fkh2:TAP; Clr6:13XMYC), or PAS837 (Fkh2:13XMYC; Sds3:TAP) were grown in 50ml YES to OD~ 1, spun and washed with STOP buffer (150mM NaCl, 50mM NaF, 10mM EDTA and 1mM $NaN_3$) and flash frozen. Cells were resuspended in 3ml ice cold HB-300 (50mM MOPS pH7.2, 300mM NaCl, 15mM $MgCl_2$, 15mM EGTA, 60mM glycerophosphate, 0.1mM $Na_3VO_4$, 2mM DTT, 1% Triton X-100, 1mM PMSF, 200μM phenantroline, pepstatin A, leupeptin, aprotinin and 1X of EDTA-free protease-inhibitor cocktail (Roche)) and then lysed in the presence of 600μl 0.5mm zirconia beads (RPI) for 6X 1min, with 5 min rest on ice in between cycles, at the maximum setting in a bead mill homogenizer (Beadruptor-12, Omni International). The lysate was clarified by spinning at 18,000 x *g* for 20min. 500μl clarified lysate was applied to a 4–20% sucrose gradient in gradient buffer (50mM Tris-HCl pH 7.5, 50mM KCl, 1mM EDTA, 1mM DTT, 1mM PMSF, 200μM phenantroline, pepstatin A, leupeptin, aprotinin and 1X of EDTA-free protease-inhibitor cocktail

(Roche)) and spun for 20hrs at 151,000 x $g$ ($r_{av}$) at 4˚C in a swinging bucket rotor (Beckman SW-41 Ti). 12x 1ml fractions were collected from the gradient and incubated for 10min at RT with 100µl 0.15% deoxycholine. Proteins were then precipitated by addition of 100µl of 50% Trichloroacetic acid and incubation on ice for 30min. Precipitates were collected by centrifugation at 16,000 x $g$ at 4˚C for 10min and pellets washed twice in ice cold acetone and then resuspended in 2X SDS-laemmli buffer and proteins separated on 1 10% SDS-PAGE gel.

## Co-immunoprecipitation

100x10^6 PAS 835 (Fkh2:13XMYC; Clr6:TAP) or PAS 837 (Fkh2:13XMYC; Sds3:TAP) cells were grown, lysed and the lysate clarified as above (sucrose gradient analysis) but in the following lysis buffer: 20mM HEPES pH7.6, 150mM NaCl, 1mM EDTA, 0.5% IGEPAL NP-40, 10% glycerol, 1mM PMSF, 200µM phenantroline, pepstatin A, leupeptin, aprotinin and 1X of EDTA-free protease-inhibitor cocktail (Roche)). 500µL total protein was incubated with 20µL IgG-Sepharose 6 resin (GE healthcare) for 2 hrs at 4˚C with rotation. Beads were washed 4 times with lysis buffer with 350mM NaCl instead of 150mM. Proteins were eluted off washed beads in 20µL 2X SDS-laemmli buffer, separated on SDS-PAGE gels and blotted as below.

## Western blot analysis

Proteins were transferred to low-fluorescence PVDF membranes (Bio-rad) at 90min at 200mA at 4˚C. Membranes were blocked in 1:1 mixture of 1XPBS: Intercept PBS blocking buffer (LiCor) and then incubated with anti-PAP (Sigma, P1291, lot 92557) or anti-MYC (Biolegend, 626802, lot B274036) at 1:1,000 either overnight at 4˚C or 90 min at RT. Membranes were washed 4X in the presence of 0.2% Tween-20 and then incubated with fluorescent anti-mouse (800nm, Rockland, 610-145-003, lot 34206) and anti-rabbit (680nm, Cell Signaling, 5366P, lot 9) at 1:5,000 and 1:15,000 respectively for 1hr at RT. Membranes were washed 4X as above, transferred to 1X PBS and imaged on a LiCor Odyssey CLx imager.

## Supporting information

**S1 Fig. Screen of chromatin contexts with "green" and "orange" reporters.** To-scale diagrams of the heterochromatin spreading sensors (without the euchromatically placed "red" reporter) in the 4 chromatin contexts used for the spreading screen (as in Greenstein et al 2018). The direction in which spreading is analyzed ("green" to "orange") is indicated per chromatin context. **A.** *WT MAT* and *MAT ΔREIII*. These two contexts are similar, except that *MAT ΔREIII* contains two short 7bp deletions of the two Atf1/Pcr1 DNA binding sites near *REIII*, inactivating it. The first binding site is not included in *REIII*, per the definitions of [98] and [23]. **B.** *MAT ΔcenH*. **C.** *ECT*. In Greenstein et al 2018, the distance between "green" and "orange" was varied for B. and C. contexts, without changes to the qualitative behavior of spreading.
(PDF)

**S2 Fig. Regulators of heterochromatin nucleation-distal silencing in all four chromatin contexts. A.** *WT-MAT* 2D-density hexbin plots of the wild-type parent, a strong heterochromatin loss hit (*Δclr3*), and the top loss of spreading hit (*Δfkh2*) in this chromatin context. Dashed blue lines indicate the values for repressed fluorescence state and dashed red lines indicate values for fully expressed fluorescence state. **B.** Beeswarm plots of Grid$_{3+4}$^mut/par for *WT MAT* loss of spreading hits. The top 10 hits are all annotated, and below those hits, mutants that show overlap with at least 3 other chromatin contexts are additionally annotated. Red line, 2SD above the Grid$_{3+4}$^mut/par of the wild-type parent isolates (black dots); dashed brown line,

the 85[th] percentile; Dot color, number of chromatin contexts with loss of spreading phenotype over the cutoff. **C.-D.** Data for the *WT ΔcenH* strain were analyzed and displayed as in E. and F. This mutation allows examination of only the *REIII* nucleation site at the *MAT* locus. **E.-F.** Data for the *WT ΔREIII* strain were analyzed and displayed as in E. and F expect that Grid$_4$^mut/par was used as the metric. **G.-H.** Data for the *ECT* strain were analyzed and displayed as in E. and F.**I.** Upset plots indicating the frequency of "loss of spreading" gene hits appearing in one or multiple chromatin contexts. For each bar, the chromatin context(s) with shared phenotypes for the underlying gene hits is indicated below the plot. The inset indicates the total number gene hits for loss of spreading in each chromatin context. "Shared genes": number of genes that appear as "loss of spreading" hits across the number of indicated chromatin contexts. (PDF)

**S3 Fig. *apm3* and *apl5*, coding for nuclear-cytosolic and cytosolic proteins, respectively, act together in modulation of heterochromatin spreading. A.-D.** 2D density hexbin plots of *de novo* generated *Δapm3* (**B.**), *Δapl5* (**C.**), and *Δapm3Δapl5* double mutant (**D.**) compared to the wild-type *MAT ΔREIII* parent (**A.**). The Fold change of Grid$_4$^mut/par is indicated in the plot. At least 3 independent isolates of each genotype are combined in each plot. **E.** Apm3:SFGFP is distributed in the cytosol and nucleus. Apm3:SFGFP was expressed from its native locus and co-expressed with Swi6:E2C. Swi6:E2C labels nuclear heterochromatin. Z-projection overlays of the Apm3:SFGFP and Swi6:E2C on top, and a brightfield image on the bottom. **F.** Apl5:SFGFP is largely nuclear excluded. Apl5:SFGFP was expressed from its native locus and co-expressed with Swi6:E2C. Swi6:E2C labels nuclear heterochromatin. Z-projection overlays of the Apl5:SFGFP and Swi6:E2C on top, and a brightfield image on the bottom. **G.** *Δapm3* exhibits a mild defect in H3K9me2 accumulation at heterochromatin islands. H3K9me2 ChIP-qPCR in wild-type parent *MAT ΔREIII* or *Δapm3* mutant. Error bars represent 1SD of three replicates. (PDF)

**S4 Fig. Gain of nucleation-distal gene silencing mutants in *WT MAT*, *MAT ΔREIII* and *ECT* chromatin contexts. A.** Beeswarm plots of Grid$_1$^mut/par for *WT MAT* gain of nucleation-distal silencing hits. The top 10 hits are all annotated, and below those hits, mutants that show overlap with 2 other chromatin contexts are additionally annotated. Red line, 2SD above the Grid$_1$^mut/par of wild-type parent isolates (black dots); dashed brown line, the 85[th] percentile; Dot color, number of chromatin contexts with loss of spreading phenotype over the cutoff. **B**. *WT MAT* 2D-density hexbin plots of the wild-type parent, and the two top gain of nucleation-distal silencing hits of this chromatin context. Dashed blue lines indicate the values for repressed fluorescence state and dashed red lines indicate values for fully expressed fluorescence state. **C.-D.** As in A., B. but for *MAT ΔREIII*. **E.-F.** As in A., B. but for *ECT*. **H.** Upset plots indicating the frequency of gain of nucleation-distal silencing gene hits that appear in the chromatin the three in contexts as in Fig 2B. For each bar, the chromatin context(s) with shared phenotypes for the underlying gene hits is indicated below the plot. The inset indicates the total number gene hits in each chromatin context of the same phenotype. (PDF)

**S5 Fig. RT-qPCR validations of selected chromatin-context unique loss and gain of nucleation-distal silencing hits.** 5 moderate- to strong hits in the loss and gain of distal silencing category that are partially or fully chromatin context-specific were selected for validations: *saf5* (gain of silencing in *WT MAT*, and moderately in *MAT ΔREIII*), *eaf6* (gain of silencing only in *MAT ΔREIII*), *pht1* and *hip1* (loss of silencing only in *ECT*), and *gad8* (strong loss of silencing in *MAT ΔcenH* and mildly in *ECT*). RT-qPCRs for SF-GFP ("*green*"-nucleation) and mKO2 ("*orange*", distal) transcripts normalized to the *act1* transcript and scaled to the wild-type

(dashed brown line, = 1) are shown for examples of: Gain of distal silencing; **A.** *WT-MAT*, **B.** *MAT ΔREIII*. Loss of distal silencing; **C.** *ECT*, **D.** *MAT ΔcenH*. Error bars indicate 1SD of 2 biological replicates generated independently from the screen. Dotted lines represent wild-type control. Note we could not recover *Δsaf5* mutants in *ECT*.
(PDF)

**S6 Fig. *gcn5* is specifically required for H3K9me2 spreading at the *ECT* heterochromatin spreading sensor, but not pericentromeric heterochromatin.** *act1*-normalized ChIP-qPCR for H3K9me2 in *ECT* wild-type parent or the *de novo* generated *Δgcn5* mutant at **A.** the pericentromeric *dg* element, and **B.** the heterochromatin spreading sensor at the *ura4* locus in *ECT*. Dumbbells indicate qPCR amplicons. Error bars indicate 1SD of 3 biological replicates.
(PDF)

**S7 Fig. Class III HDAC family Sir2 is required for heterochromatin silencing.** 2D density hexbin plots of *Δsir2* mutants in each chromatin context from the screen. Mutation in *sir2* causes a loss of silencing phenotype in all examined chromatin context.
(PDF)

**S8 Fig. 2D density hexbin plots for all Clr6 complex subunit screen mutants in *MAT ΔREIII*.** 2D density hexbin plots of all Clr6 complexes gene mutants from the screen, corresponding to Fig 4E in *MAT ΔREIII* context. The mutants are arranged in descending order of $\text{Grid}_{3+4}^{\text{mut/par}}$; in *MAT ΔREIII* only *Δfkh2*, *Δcph1*, *Δpng2*, *Δdep1*, *Δprw1* and *Δlaf1* were identified as loss of spreading phenotype. Original *MAT ΔREIII* wild type parent and mutants shown in Figs 1 and 4E are shown here again (with transparency) for comparison. GO complex annotations are indicated next to each mutant by colored boxes.
(PDF)

**S9 Fig. Fkh2- containing Clr6 complexes direct H3K9me2 spreading at multiple genomic regions. A.** Principal Component Analysis was performed on the normalized counts in the top 500 most variable bins (analogous to RNA-Seq analysis, TOP) or all 1500bp bins (BOTTOM) passing a threshold for global enrichment of H3K9me2 signal (see Materials and Methods). The first two principal component values are plotted for each sample with genotypes as defined in the legend. **B.-F.** Signal tracks plots for the MAT locus and indicated centromeres and telomeres as in the main text. No nucleator sequences are present on subtelomere IR so the first annotation row below the signal tracks is empty.
(PDF)

**S10 Fig. Fkh2 and Prw1 act together in spreading H3K9me2. A.** H3K9me2 ChIP-qPCR at the MAT locus in wild-type *MAT ΔREIII*, *Δfkh2*, *Δprw1*, and the *Δfkh2Δprw1* double mutant. **B.** As in A., at indicated heterochromatin islands. **C.** As in A., at *tel IL*. Error bars represent 1SD of 3 biological replicates.
(PDF)

**S11 Fig. Effect of *clr6-1* and *Δprw1* on H3K9me2 at a WT MAT locus without reporters.** H3K9me2 ChIP-qPCR at the MAT locus in a wild-type MAT locus (no heterochromatin spreading reporters, see diagram), *Δprw1*, and *clr6-1*, at indicated amplicons (dumbbells). Error bars represent 1SD of 3 biological replicates.
(PDF)

**S12 Fig. Fkh2- containing Clr6 complexes contribute primarily to H3K9me2 spreading, while Clr3 is required for H3K9me2 accumulation at all heterochromatin regions except islands. A.-D.** Volcano plots representing -log10(adjusted p-value) vs log2FoldChange values

for mutants (I., *Δfkh2;* J., *Δprw1;* K, *clr6-1;* L., *clr3-D232N*) over WT. P-values were corrected for multiple testing with the Benjamini-Hochberg procedure. Cutoff values for adjusted p-value < 0.005 and absolute value Log2FoldChange > 1 are annotated on the plot. Dots represent individual 300bp windows tested for differential enrichment. Dots are colored by their annotation to nucleation or spreading zones, presence within a previously identified euchromatin embedded H3K9me2 heterochromatin region ("island", "HOOD", or "region"), or regions outside these categories (other). **E.-H.** Volcano plots were generated as in A-D. Dots are colored by their annotation to nucleation or spreading zones broken down by heterochromatin location (pericentromere, subtelomere, MAT) or presence within a previously identified euchromatin embedded H3K9me2 heterochromatin region. **I.** The number of regions called as significant in each direction for each of the pairwise comparisons is tabulated per each category of genomic feature. **J.** The overlap of regions identified as significantly reduced in H3K9me2 signal in each mutant vs WT is compared in a Venn Diagram.
(PDF)

**S13 Fig. Clr6 affects spreading of H3K9me3. A.** H3K9me3 ChIP-qPCR at the MAT locus in wild-type *MAT ΔREIII*, *Δfkh2*, and *clr6-1*. **B.** As in A., at *dg* repeats, which are at the distal end of the left of the pericentromere at *cen II* and an amplicon 2.5kb beyond the last annotated nucleating feature at *cen II* left. **C.** As in A., but at heterochromatin islands *mei4* and *mcp7*. **D.** As in A., but at *tel 1L*. Error bars represent 1SD of 3 biological replicates.
(PDF)

**S14 Fig. Fkh2 is a constituent member of Clr6 complexes. A.** Co-Immunoprecipitation experiment with baits Clr6-TAP or Sds3-TAP and prey Fkh2-MYC. LEFT: Western blot against indicated proteins for the Co-IP experiment. RIGHT shows the entire western blot, including lanes unrelated to the co-IP experiment (2–4,5,10,12). **B.** Sucrose density gradient for whole cell extracts of cells containing Sds3-TAP, a signature of complex I/I″, and Fkh2:MYC. Gradient and Western as in Fig 6. **C.** Single channel Western blots of Fig 6A.
(PDF)

**S15 Fig. Fkh2 does not affect transcription of core heterochromatin regulators.** RT-qPCR of indicated core heterochromatin regulators, including representatives of ClrC, Clr6, and SHREC in *MAT ΔREIII* wild-type or *Δfkh2* cells. Heterochromatin regulator transcripts are normalized to the *act1* transcript and shown on a log2 scale, given the wide distribution of transcript abundance between indicated regulators. Error bars indicate 1SD of 2 biological replicates.
(PDF)

**S1 Table. Nuclear function gene deletion library.** List of gene deletion strains used for genetic screens in this study.
(PDF)

**S2 Table. Strain table.** List of *S. pombe* strains used in this study.
(PDF)

**S3 Table. Primers used for ChIP qPCR and RT qPCR.** Primers for amplicons used in qPCR in this study.
(PDF)

## Acknowledgments

We thank the Michael N Boddy lab for their generous gift of strains expressing tagged Clr6 I″ complex subunits. We thank Sy E Redding, Carol A Gross, Douglas Myers-Turnbull and

Kamir Hiam for helpful discussions on data acquisition, analysis, and interpretation, Arthur Molines for help with microscopy experiments, and Sandra Catania and Hiten D Madhani for support for chromatin immunoprecipitation experiments.

## Author Contributions

**Conceptualization:** R. A. Greenstein, Sigurd Braun, Bassem Al-Sady.

**Data curation:** R. A. Greenstein, Henry Ng.

**Formal analysis:** R. A. Greenstein, Henry Ng.

**Funding acquisition:** Sigurd Braun, Bassem Al-Sady.

**Investigation:** R. A. Greenstein, Henry Ng, Ramon R. Barrales, Catherine Tan, Bassem Al-Sady.

**Methodology:** R. A. Greenstein, Henry Ng, Ramon R. Barrales, Catherine Tan, Bassem Al-Sady.

**Project administration:** Bassem Al-Sady.

**Resources:** Ramon R. Barrales, Sigurd Braun.

**Software:** R. A. Greenstein, Henry Ng.

**Supervision:** Catherine Tan, Sigurd Braun, Bassem Al-Sady.

**Visualization:** R. A. Greenstein, Henry Ng, Bassem Al-Sady.

**Writing – original draft:** R. A. Greenstein, Henry Ng, Bassem Al-Sady.

**Writing – review & editing:** R. A. Greenstein, Henry Ng, Ramon R. Barrales, Sigurd Braun, Bassem Al-Sady.

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
