## [Decision Letter · Decision Letter 0]

22 Nov 2021

Dear Dr Al-Sady,

Thank you very much for submitting your Research Article entitled 'Local chromatin context regulates the genetic requirements of the heterochromatin spreading reaction.' to PLOS Genetics.

The manuscript was fully evaluated at the editorial level and by independent peer reviewers. The reviewers appreciated the attention to an important topic but identified some concerns that we ask you address in a revised manuscript.

We therefore ask you to modify the manuscript according to the review recommendations. Your revisions should address the specific points made by each reviewer.

2) Please make sure that supporting datasets and code are made publicly available. We were unable to find 10.5281/zenodo.5122951. 

3) Upload a Striking Image with a corresponding caption to accompany your manuscript if one is available (either a new image or an existing one from within your manuscript). If this image is judged to be suitable, it may be featured on our website. Images should ideally be high resolution, eye-catching, single panel square images. For examples, please browse our archive. If your image is from someone other than yourself, please ensure that the artist has read and agreed to the terms and conditions of the Creative Commons Attribution License. Note: we cannot publish copyrighted images.

[LINK]

Yours sincerely,

Bas van Steensel

Associate Editor

PLOS Genetics

Wendy Bickmore

Section Editor: Epigenetics

PLOS Genetics

Reviewer's Responses to Questions

**Comments to the Authors:**

Reviewer #1: “Local chromatin context regulates the genetic requirements of the heterochromatin spreading reaction” by Greenstein et al.

Heterochromatin is a repressive chromatin structure that can spread across large domains to enforce stable silencing of gene-poor regions containing repetitive elements. Heterochromatin domains can also form across gene-rich regions to elicit the signature gene silencing pattern of a specific cell lineage. In this manuscript, the authors investigate how genomic context contributes to heterochromatin spreading activity by using a quantitative silencing assay in combination with a genetic screen, which allows them to uncover factors that impact heterochromatin-mediated silencing at locations distal to a nucleation site. The authors report that the factors that support spreading do indeed differ depending on genomic context. From their analyses, the authors conclude that the Clr6 deacetylase is recruited by the FKH2 transcription factor specifically at distal sites of constitutive heterochromatin formation, but not within euchromatin or near nucleation sites. By contrast, Clr3 deacetylase is more generally required for spreading from nucleation sites throughout the entire domain. The authors also report that chromatin remodelers serve to antagonize heterochromatin spreading at distal sites.

Critique: This is an interesting study. The authors have identified several factors that differentially affect the heterochromatic silencing of reporters in different chromosomal contexts. This is highly significant as the factors identified in their screen would serve as an important community resource. Although it is clear from their extensive analyses that Clr6 deacetylase complex specifically affects silencing of reporter inserted distal to the nucleation sites, the interpretation that the observed changes are due to defects in spreading of heterochromatin is not fully supported by the results (e.g. Supporting Fig 7 shows little or no change in H3K9me2 levels across the MAT). The text needs to be revised extensively to more accurately describe the results presented. Specifically, the suggestions throughout the manuscript (and in the title) that the observed effects indicate changes in heterochromatin spreading should be removed before submitting a revised version for publication in PLoS Genetics.

Specific comments:

(1) The authors suggest that Clr6-related mutants affect the spreading of heterochromatin to mediate silencing of nucleation distal reporter. Whereas it is clear that Clr6 mutants differentially affect the silencing of nucleation proximal and distal reporters, these mutants show no major defects in heterochromatin spreading (e.g. see supplementary Fig. 7B). An important possibility is that the observed changes in gene expression are due to local chromatin changes in mutant cells rather than due to broader defects in the spread of heterochromatin. This possibility is supported by changes in H3K9me2 levels specifically near the orange reporter that is derepressed in Clr6 mutants. Related to this point, it would be better if the authors mapped the distribution of H3K9me3 as the spread of heterochromatin in S. pombe has been shown to require binding of histone methyltransferase Clr4 to H3K9me3.

(2) Similarly, the authors suggest that the Clr6 related mutants affect heterochromatin spreading at centromeres. While some changes in H3K9me2 levels can be seen, it is not clear whether these changes can be characterized as spreading defects. On the other hand, changes in H3K9me2 are observed at telomeres in the Clr6 complex mutant. However, it is unclear if Clr6 is directly involved and that these changes are functionally linked to changes in gene silencing.

(3) The authors conclude that another deacetylase Clr3 is more broadly required for H3K9me2 and gene silencing, including at nucleation sites. As mentioned in the discussion, this conclusion seems to be different from previous studies that appear to have not observed any reduction in H3K9me2 at centromeric repeats in clr3∆ cells. They note that variable results might be due to a specific mutant (Clr3-D232N) used in their study as compared to clr3∆ used in previous studies. Since all functional assays (such as results of the genetic screen described in Fig. 1) were performed using clr3∆, it seems important that these results are correlated with changes in H3K9me2 in the same strain (clr3∆). Also, the Clr3-D232N cells appear to maintain significant levels of H3K9me at the nucleation sites (Fig. 5 and S7); suggesting that heterochromatin assembles at nucleation sites in clr3 mutant cells but fails to spread to distal sites. The text in the abstract and main text should be modified to more accurately describe these results.

(4) It is argued that the Clr6 complex is specifically involved in distal “heterochromatin spreading” but is not required at nucleation sites. Considering that HDACs have overlapping functions, a possibility cannot be ruled out Clr6 involvement at nucleation sites is masked by other HDACs. The text needs to be modified and arguments about Clr6 acting downstream of Clr3 shall be reconsidered.

(5) The authors suggest that Fkh2 promotes heterochromatin spreading by recruiting Clr6 to distal sites. Since Fkh2 is a DNA binding protein, it is unclear how this factor targets Clr6 to distal regions. Does Fkh2 bind to specific sequences near the distal site?

(6) Loss of H2AZ negatively affects silencing of the distal marker, suggesting that it is required for “heterochromatin spreading”. The authors wrote, “H2A.Z is known to antagonize heterochromatin spreading in budding yeast (36), indicating that the role of this histone in preventing gene silencing by chromatin regulation is conserved.” How is function conserved? In contrast to H2AZ, the loss of SWR1 complex involved in loading H2AZ has an opposite effect. This shall be explained in the text.

Reviewer #2: The primary goal of this manuscript is to identify regulatory factors that enhance or suppress heterochromatin spreading. The authors use a very elegant two-color fluorescence reporter system to uncouple nucleation from the heterochromatin spreading. The authors primarily focus on spreading at the MAT locus and at an ectopic site where a dh reporter fragment has been inserted. The manuscript provides a comprehensive accounting of non-essential proteins that enhance or suppress heterochromatin spreading. The authors performed a very elegant screen to figure out which chromatin regulators enhance or suppress heterochromatin spreading. I also liked the idea of drawing a contrast between clr6 and clr3- both being histone deacetylases but one affecting spreading and the other affecting nucleation. The data about Fkh2 being in complex with Clr6 is very clear and the data is internally consistent in identifying Fkh2 as a top hit in the genetic screen.

The quality of the data, the analysis frameworks and the very careful discussion of the results are strengths of this manuscript. Heterochromatin spreading is dynamic and I really like that the authors use a fluorescence-based approach with the sensitivity to parse out different populations of cells to investigate this process. In summary, this is a technically solid manuscript, and the screen is very well designed. However, I do have some major concerns that should be addressed. The first of my comments is a conceptual issue I have with how the screen is presented and the other issues relate to additional experiments that are needed to solidify author claims. I support publication of this manuscript provided the authors can be responsive to these concerns.

1) The authors present findings of different chromatin regulators (this is more of an issue in the loss of function factors) whose deletions lead to an increase in heterochromatin spread. Three genetic contexts in their screen involve the MAT locus. When eliminating different nucleation and/or spreading elements (such as cenH and REIII), the authors find different lists of chromatin regulators that affect spreading in their screen. It is very likely that these lists are different because each represents a sensitized background of some kind where nucleation is affected (I am aware the authors pre-selected 'green' cells but heterochromatin is dynamic and pre-selecting green cells does not mean that nucleation is not sensitized to stochastic loss). It is very important that this potential pitfall is laid out very clearly in the design of the screen. Their results imply there is a hierarchy of chromatin regulators that affect spreading- some in strong nucleation contexts and others in weak or impaired nucleation contexts (such as REIII and cenH deletion). But it is very important to present the discovery of the different chromatin regulators in a unified manner as it concerns the MAT locus. In my view, the authors are only comparing two loci- MAT and ECT and all other strains are a subset of MAT.

2) An extension of the previous comment is that figures such as the upset plots in my opinion become complicated by including REIII and cenH deletion strains as genetic contexts that are separate from MAT. Furthermore, presenting hits of their screen in the REIII and cenH deletion backgrounds as part of the main figures makes the manuscript feel a bit more tedious. However, I want to be clear that what the authors choose to do about their figures has no consequence for my review. This comment is merely to be treated as one person’s opinion of an area where I think the manuscript can benefit in terms of clarity.

3) In Figure 5D, the authors present H3K9me2 ChIP-seq data. The PCA analysis and ChIP-seq tracks of clr6-1 clearly suggest that the H3K9me2 patterns look different from that of wild-type cells. However, in many cases H3K9me2 levels in clr6-1 strains are higher than that of wild-type strains. This is a mechanistic detail that is important to a key idea in the manuscript. The authors should address this concern by including ChIP-qPCR H3K9me3 data (ChIP-seq is also fine) to determine if changes in H3K9me3 could explain the spreading defects that are unique to the Clr6 hypomorph. Without the H3K9me3 data (regardless of the outcome itself) it is unclear what the molecular nature of the spreading defect actually is!

4) The authors use two fluorescent reporters to visualize changes in gene silencing. The decrease in H3K9me2 levels in clr6-1 strains at the site of the orange reporter insertion is concerning (Figure 5D and related interpretations in Figure 5). Since the manuscript focuses on how local chromatin context changes affect spreading, the authors must test how insertion of their reporter is affecting spreading especially in some of their mutant contexts (clr6-1 should be sufficient). My recommendation for an experiment would be an H3K9me2 ChIP-qPCR measurement in a wild-type and clr6-1 strain where the orange reporter has NOT been inserted. Directly comparing the two results will reveal the extent to which the reporter itself affects the observation. This does not detract from the strength of the manuscript but will provide a note of caution for other ways in which their results can be interpreted.

5) Regarding Fkh2, its DNA binding properties and mechanism of regulating heterochromatin spreading (Figure 6): If the authors are invested in the claim that Fkh2 acts independently of its DNA binding domain then its important they either mutate or delete the DNA binding domain. In the absence of including such data, the authors need to modify their interpretations. While they have assayed a handful of heterochromatin proteins for their mRNA levels, it is impossible to rule out there may be other factors (expected or unexpected perhaps) that are regulated by Fkh2. Making the relevant Fkh2 mutants (either DBD mutant or a point mutant) is essential to support their claims made in Figure 6 and the discussion section.

Reviewer #3: Silent chromatin structure, heterochromatin can spread from the nucleation site. Fission yeast provides a good model to analyze heterochromatin formation and many research revealed that several nucleation systems and many factors for nucleation has been identified. But the mechanism and the factors for spreading are not well understood. Greenstein et al. tried to clarify this issue using their single cell analysis system that enable to quantitative measurement of the nucleation and the spreading of the silent state. Screening of deletion library revealed that specific genes are required for the spreading and the same different genes promote spreading from different nucleation sites.

This manuscript provides comprehensive list of genes required for spreading of silent state and suggests context-dependent spreading mechanisms. Though the quality of the data is high, the results are rather descriptive and lacking the detail of the mechanisms except the analysis of Fkh2-containing Clr6 HDAC complex. However, I think that the information in this manuscript beneficial for many researchers working in chromatin and/or epigenetic field and suitable for publishing in PLOS Genetics.

Minor points

Supporting Figure 1B: MAT�K should be MAT�cenH

Figure 4 E-G and Supporting Figure 7 B-D: Fission yeast centromeres basically have symmetric repeats structure, but in the figures, allows, which (I suppose) represent repeats, are not symmetric.

**Have all data underlying the figures and results presented in the manuscript been provided?**

Reviewer #1: Yes

Reviewer #2: Yes

Reviewer #3: Yes

PLOS authors have the option to publish the peer review history of their article (what does this mean?). If published, this will include your full peer review and any attached files.

Reviewer #1: No

Reviewer #2: No

Reviewer #3: **Yes: **Yota Murakami

---

## [Decision Letter · Decision Letter 1]

13 Apr 2022

Dear Dr Al-Sady,

We are pleased to inform you that your manuscript entitled "Local chromatin context regulates the genetic requirements of the heterochromatin spreading reaction." has been editorially accepted for publication in PLOS Genetics. Congratulations!

Yours sincerely,

Bas van Steensel

Associate Editor

PLOS Genetics

Wendy Bickmore

Section Editor: Epigenetics

PLOS Genetics

Comments from the reviewers (if applicable):

Reviewer's Responses to Questions

**Comments to the Authors:**

Reviewer #1: I have read the revised version of the manuscript. The authors have addressed my main concerns and manuscript can be now considered for publication in PloS Genetics.

Reviewer #2: The response of the authors to the questions raised during review are satisfactory. In particular, I want to highlight their H3K9me3 ChIP experiments which I really appreciate and provides strong support for their model of how different chromatin factors affect spreading and silencing. I have do not have any further concerns about the manuscript.

Minor edit that I noticed while reading the authors response: Line 363 - K3K9me2 needs to be H3K9me2.

**Have all data underlying the figures and results presented in the manuscript been provided?**

Reviewer #1: Yes

Reviewer #2: Yes

PLOS authors have the option to publish the peer review history of their article (what does this mean?). If published, this will include your full peer review and any attached files.

Reviewer #1: No

Reviewer #2: No

**Data Deposition**

http://datadryad.org/submit?journalID=pgenetics&manu=PGENETICS-D-21-01401R1

**Press Queries**

---

## [Editor Report · Acceptance letter]

10 May 2022

PGENETICS-D-21-01401R1 

Local chromatin context regulates the genetic requirements of the heterochromatin spreading reaction 

Dear Dr Al-Sady, 

We are pleased to inform you that your manuscript entitled "Local chromatin context regulates the genetic requirements of the heterochromatin spreading reaction" has been formally accepted for publication in PLOS Genetics! Your manuscript is now with our production department and you will be notified of the publication date in due course.

With kind regards,

Livia Horvath

PLOS Genetics

On behalf of:
